# MIMIC-BENCH: EXPLORING THE USER-LIKE THINKING AND MIMICKING CAPABILITIES OF MULTIMODAL LARGE LANGUAGE MODELS

**Jiajie Teng**[1], **Huiyu Duan**[1,*] **Sijing Wu**[1], **Jiarui Wang**[1], **Xilei Zhu**[1], **Jianing Jin**[1],
**Wei Shen**[1], **Xiongkuo Min**[1], **Guangtao Zhai**[1]
[1]Shanghai Jiao Tong University

## ABSTRACT

The rapid advancement of multimodal large language models (MLLMs) has greatly prompted the video interpretation task, and numerous works have been proposed to explore and benchmark the cognition and basic visual reasoning capabilities of MLLMs. However, practical applications on social media platforms demand MLLMs that can emulate user-like thinking and behavior when interpreting user-generated videos, which has been rarely studied in current research. To bridge the gap and get closer to general practical artificial intelligence (AI), we first construct **MIMIC-Data**, a large-scale dataset containing 150K+ user-shared videos with corresponding information including captions, tags, comments, *etc.* Then, we present **MIMIC-Bench**, a large-scale benchmark building upon curated 4,000 user-shared videos from MIMIC-Data, which is designed to evaluate user-like thinking and mimicking capabilities of MLLMs in real-world video contexts. MIMIC-Bench not only supports user-like thinking challenges including creator intent, user feedback interpretation, *etc.*, but also introduces a novel comment imitation task to assess whether MLLMs can generate human-like responses to video content. Based on MIMIC-Data and MIMIC-Bench, we develop **MIMIC-Chat**, which integrates spatial and temporal features into a large language model, and finetunes the model to perform user-like thinking and mimicking tasks. Extensive experiments conducted based on 24 existing MLLMs and our MIMIC-Chat model that current MLLMs exhibit limited capabilities to perform human-like thinking and responses, and MIMIC-Chat performs better to some extent. We hope MIMIC-Bench can contribute to the advancement of human-aligned video understanding in the multi-modal era. The MIMIC-Data, MIMIC-Bench, and MIMIC-Chat will be released upon the publication.

## 1 INTRODUCTION

In recent years, video platforms have rapidly emerged as mainstream media for social interaction and knowledge sharing. Beyond passive viewing, users actively participate in shaping the video ecosystem through content creation, commenting, and engagement. A single user-generated video is often accompanied by rich metadata and viewer interactions, such as titles, descriptions, tags, and comments, *etc.*, which reflect not only the video content but also how humans perceive, interpret, and respond to it. These contents embody high-level human cognition and social behavior, offering a rich yet underexplored resource for evaluating machine intelligence. Studying how multi-modal large language models (MLLMs) interpret such human-centric signals is crucial for advancing toward general-purpose AI systems that can truly understand and engage into social and cultural contexts.

Driven by the effectiveness of integrating multimodal information into large language models (LLMs) to acquire the scene perception ability (Radford et al., 2021; Li et al., 2021; Chen et al., 2020; Wang et al., 2022; Cho et al., 2021; Wang et al., 2021), recent progress in MLLMs (Zhang et al., 2025; Bai et al., 2025; Gao et al., 2024; Chen et al., 2024a) have led to strong performance on tasks involving visual understanding, temporal reasoning, and open-ended language generation. Powered by scalable visual encoders and multi-frame alignment mechanisms, these models achieve impressive results on several benchmarks (Li et al., 2024b; Fu et al., 2024; Mangalam et al., 2023), *etc.*, which assess the capabilities of MLLMs in video captioning, action recognition, and event prediction. However, these benchmarks typically rely on curated visual-only datasets and designer-

---

*Corresponding author.

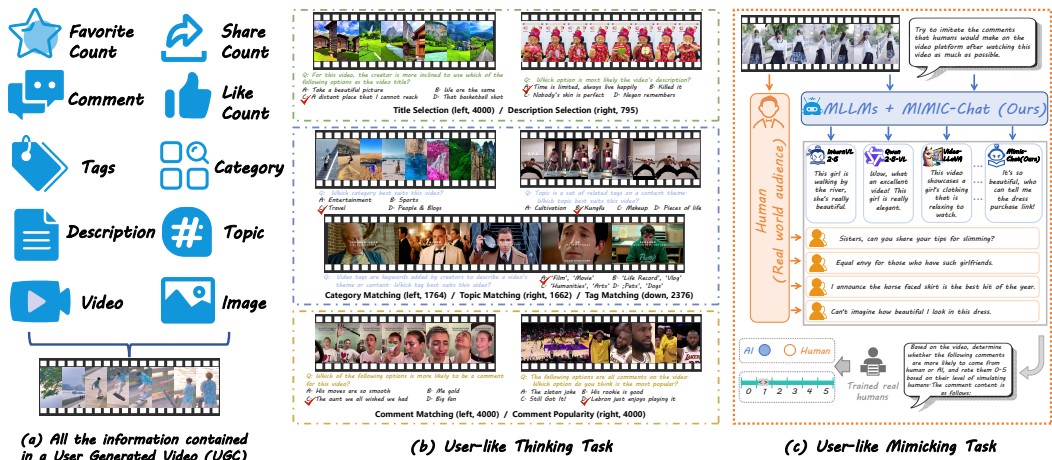

Figure 1: Overview of **MIMIC-Bench**. **(a)** Left: Multi-source metadata from user-shared videos. **(b)** Center: **User-like thinking task**: across three axes—CIU, CAM, and UIU. **(c)** Right: **User-like mimicking task**: human-like comment simulation pipeline.

crafted questions, focusing on low- or mid-level perception tasks that emphasize what happens in the video while neglecting how humans think , feel, and react. The lack of real-world user context and organically generated content limits their capacity to evaluate whether MLLMs can simulate human cognition and language behavior in authentic video scenarios. Although some studies have begun to explore emotions in multimodal data such as EmoLLM (Yang et al., 2024), *etc.*, the practical applications are more complex than simple emotional tasks.

To bridge this gap, we first introduce **MIMIC-Data**, which comprises 150K+ user-shared videos and corresponding metadata information, as shown in Figure 1(a). Then, we construct **MIMIC-Bench**, a large-scale benchmark designed to evaluate user-like thinking and behavioral capabilities of MLLMs in real-world, user-centric video applications. Unlike prior benchmarks that focus primarily on visual recognition and rely on synthetic or designer-curated questions, MIMIC-Bench is grounded in real user-generated content and targets the human-like video understanding, *i.e.*, how creators and viewers interpret, react to, and communicate about video content. As shown in Figure 1(b)(c), the benchmark consists of two components: (1) a **User-like Thinking Task** that spans three structured reasoning axes, including Creator Intent Understanding (CIU), Content Attribute Matching (CAM), and User Interaction Understanding (UIU), and (2) a **User-like Mimicking Task** that evaluates whether MLLMs can generate and identify human-like comments on video content. Notably, MIMIC-Bench shifts the focus from factual QA to human-centered cognitive tasks, especially comment imitation, which directly evaluate whether MLLMs can approximate human-like reasoning and expression.

We also introduce **MIMIC-Chat**, a multi-modal model designed to simulate human thinking and expression. Built upon the spatial and temporal visual encoders and a large language model backbone, it is trained on over 150k video samples from the training set of MIMIC-Data, enabling joint learning of video semantics and human-style responses. In summary, our key contributions are:

- We construct **MIMIC-Data**, a large-scale dataset containing all metadata information for over 150K user-shared videos.

- We propose **MIMIC-Bench**, the first large-scale benchmark designed to evaluate human-aligned reasoning and communicative behavior of MLLMs in practical video understanding. Especially, we introduce a novel **comment imitation task** that offers a new lens to assess human-like thinking and mimicking capabilities of MLLMs.

- We develop **MIMIC-Chat**, a multi-modal model trained on 150k+ real-world samples, enabling the joint modeling of video semantics and social cognition.

- We conduct extensive experiments on MIMIC-Bench, and compare the performance of 24 state-of-the-art MLLMs and our MIMIC-Chat. Results show that current MLLMs exhibit limited capabilities on MIMIC-Bench, and MIMIC-Chat performs better to some extent.

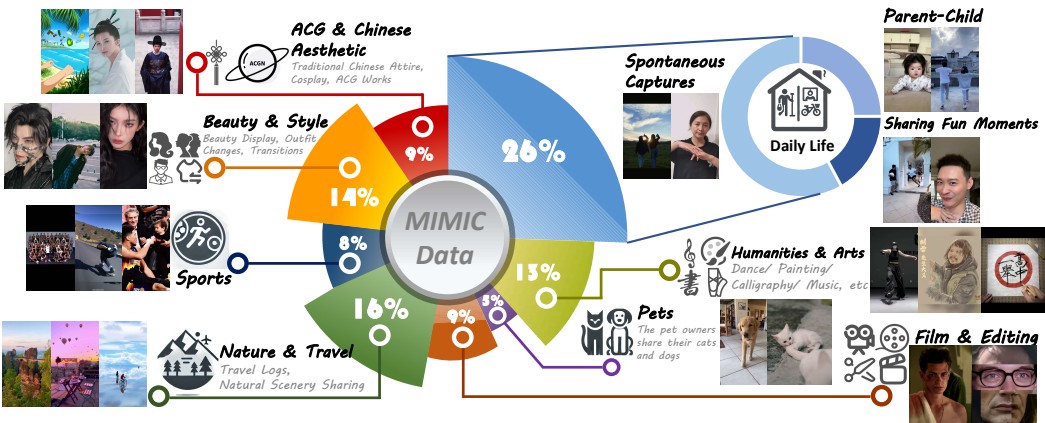

Figure 2: **Mimic-Data Overview**. We select 150,000+ videos, and each video is guaranteed to be of high quality and in line with public aesthetics. The videos can be categorized into 8 categories (Daily Life can be divided into: Spontaneous Captures, Parent-Child, and Sharing Fun Moments).

## 2 RELATED WORK

### 2.1 MULTI-MODAL LARGE LANGUAGE MODELS FOR VIDEO UNDERSTANDING

Recent advances in large language models (LLMs) (Eysenbach et al., 2023; Chiang et al., 2023) have shown strong performance in open-ended reasoning, contextual understanding, and instruction following (Radford et al., 2019; Brown et al., 2020; Ouyang et al., 2022; Touvron et al., 2023). Building on this progress, multi-modal large language models (MLLMs) (Dai et al., 2023; OpenAI, 2023; Team et al., 2023; Zhu et al., 2023) have emerged to jointly process visual and textual inputs. Early works (Alayrac et al., 2022; Chen et al., 2022) introduced vision-language pretraining, while recent models (Liu et al., 2023; Sun et al., 2024) improved image-language alignment. As video becomes more prominent, MLLMs have been extended to handle temporal dynamics and multi-frame reasoning. Representative models like Video-LLaMA (Zhang et al., 2023; Cheng et al., 2024; Zhang et al., 2025), InternVL (Chen et al., 2023; 2024b; Gao et al., 2024; Chen et al., 2024a), and the Qwen-VL series (Bai et al., 2023; Wang et al., 2024a; Bai et al., 2025) combine visual encoders with language backbones to perform well on video captioning, action localization, and Q&A, leveraging spatiotemporal pooling, contrastive learning, and dense alignment. Despite excelling at factual understanding, most MLLMs fall short in modeling human-like cognition. These gaps highlight the need reflecting real-world user intent and communication and motivates our work.

### 2.2 VIDEO BENCHMARKS AND EVALUATION PROTOCOLS

Numerous benchmarks (Fu et al., 2023; Li et al., 2023; Yu et al., 2023; Liu et al., 2025; Ning et al., 2023; Wang et al., 2024b) and datasets (Marino et al., 2019; Goyal et al., 2017; Song et al., 2024; Duan et al., 2025; Xu et al., 2025; Wang et al., 2025a; Yang et al., 2025) have been developed to evaluate MLLMs on video understanding. Early works (Jang et al., 2017; Chowdhury et al., 2018) focus on short-form video QA, testing recognition of visual entities and actions. Recent benchmarks (Li et al., 2024b; Fu et al., 2024; Mangalam et al., 2023; Wu et al., 2024) extend evaluation to multimodal alignment, instruction following, long-range modeling, and multi-turn dialogue. These resources have improved content-level evaluation, primarily assessing "what happens in the video" via structured queries, while overlooking how humans interpret, react to, or linguistically engage with videos. Most evaluations stress visual recognition but neglect emotional and linguistic engagement with videos, while EmoLLM (Yang et al., 2024) touches affective reasoning yet leaves human-style commenting, social interaction, and expressive realism underexplored. User-generated signals like comments, descriptions, and feedback are rarely included, despite their relevance on real-world platforms. This limits assessment of models' ability to simulate human cognition and communication in natural contexts. In contrast, our benchmark incorporates real user content and authentic responses to evaluate MLLMs from a human-aligned perspective.

## 3 BENCHMARK CONSTRUCTION – MIMIC-BENCH

In this section, we introduce **MIMIC-Bench**, a benchmark for evaluating human-aligned reasoning and communicative behavior in real-world, user-centric video scenarios. It follows three core principles: **Realism**—deriving tasks from authentic human interactions rather than synthetic prompts; **Task Orientation**—designing evaluations around multiple-choice and generation tasks with prac-

tical grounding; and **Human Alignment**—measuring models' ability to simulate human cognition and expression in the video domain.

## 3.1 DATASET OVERVIEW

**Data Source and Initial Collection.** To construct MIMIC-Bench, we first collect over 150,000 user-generated videos from mainstream short video platforms (TikTok and YouTube), which form the foundation of **MIMIC-Data**. This dataset spans diverse topics such as lifestyle, education, travel, beauty, and humor (Figure 2). Each video is paired with metadata, including titles, tags, descriptions, comments, and interaction statistics, all sourced from publicly accessible content in compliance with platform policies. To ensure quality and usability, we applied initial filtering to remove videos with low resolution, excessive noise, or corruption, as well as those lacking meaningful metadata (*e.g.*, empty titles, no comments, very short duration). We prioritized samples with rich viewer engagement—measured by comment volume and like count—ensuring that selected videos reflect natural and diverse human responses. This high-quality pool forms the basis for both benchmark construction and model training.

We further examine the video duration distribution of **MIMIC-Data**, which serves as the source pool for benchmark construction. Specifically, 32.76% of the videos are between 1–10 seconds, 49.08% fall within 10–45 seconds, 11.05% span 45–180 seconds, and 7.11% exceed 180 seconds in length. While MIMIC-Data predominantly consists of short-form videos, it also contains a nontrivial proportion of medium- and long-duration content. This diversity in video length ensures that the subsequent benchmark subset is grounded in a realistic distribution of user-generated video scenarios, rather than being restricted to narrowly defined short clips.

**Multi-Source Metadata Composition.** Each video is represented as a multi-modal, multi-field unit combining content with surrounding human interactions. Beyond the visual and audio signals, each sample includes user-level metadata: title (creator intent), tags and topic labels (categorical annotations), description (context or emotional framing), category, user comments (subjective impressions), and popularity indicators (like counts). These components capture not only what the video presents but also how creators frame it and how viewers respond cognitively and emotionally. Unlike datasets focused solely on video-text pairs, our structure reflects the full communication loop of online video ecosystems.

**Benchmark Subset Construction.** To build tasks, we further filtered the 150,000 videos using a ranking score based on engagement metrics (likes, favorites, shares), publisher influence, and thematic relevance. We selected the **top 2%** from TikTok and **top 5%** from YouTube, yielding 4,000 highly engaging and representative samples.

These videos combine high content quality with rich user interaction and serve as the core resource for constructing challenging, cognitively demanding evaluation tasks.

To avoid data leakage and memorization, MIMIC-Bench was constructed *prior* to model training through the above multi-stage filtering process. All benchmark videos are strictly excluded from the training set, and the remaining non-overlapping videos constitute the final MIMIC-Data used for training. As a result, MIMIC-Bench and MIMIC-Data are disjoint at the video level and serve distinct purposes, with the former used exclusively for evaluation and the latter for large-scale training.

To ensure reliability, we implemented a three-step quality control process: selecting highly interactive videos with rich viewer feedback, carefully designing distractors to avoid ambiguity, and conducting a large-scale manual review of over 20,000 QA entries from 4,000 videos. This procedure guarantees accuracy, coherence, and unambiguous semantics across the benchmark. The distribution of tasks reflects real-world user behavior rather than being artificially balanced, ensuring that the benchmark maintains ecological validity and mirrors authentic application scenarios.

**Fairness and Validity** To enhance fairness and temporal validity, we further filtered out samples that relied on ephemeral trends, niche memes, or culture-specific references. By retaining only general and widely comprehensible semantics—such as emotional tone, social context, and stylistic cues—MIMIC-Bench remains accessible across cultural backgrounds and stable over time.

All data collection strictly follows the copyright policies and terms of service of the respective platforms. We do not redistribute raw video files and only release benchmark annotations and metadata necessary for academic research. Additional details on copyright compliance, de-identification procedures, exclusion criteria, and risk mitigation protocols are provided in Appendix F.

## 3.2 TASK SUITE DESIGN

### 3.2.1 USER-LIKE THINKING TASK

To assess whether MLLMs can reason about video content in a human-aligned way, we design seven single-choice tasks grouped into three axes: Creator Intent Understanding (CIU), Content Attribute Matching (CAM), and User Interaction Understanding (UIU). These tasks leverage rich metadata from **MIMIC-Data** to evaluate both perceptual and cognitive understanding. These tasks are grounded in rich video metadata and evaluate both perceptual and cognitive understanding. Detailed information about the MIMIC-Bench benchmark, are provided in Appendix.

**Creator Intent Understanding (CIU).** This axis includes two sub-tasks. In *Title Selection*, the model selects the title most likely assigned by the creator, based on thematic and stylistic cues. In *Description Selection*, it identifies the description best aligned with the video's content and intent. Ground-truths come from original metadata; distractors are sampled from unrelated videos to ensure semantic contrast and minimize stylistic leakage.

**Content Attribute Matching (CAM).** This axis comprises three tasks. In *Tag Matching*, the model selects the tag that best captures content-level semantics. *Topic Matching* requires identifying the most relevant thematic label, while *Category Matching* asks for the correct predefined content category. All answers are derived from original annotations, and distractors are drawn from distinct videos with non-overlapping labels.

**User Interaction Understanding (UIU).** This axis focuses on modeling viewer behavior. In *Comment Matching*, the model selects the comment most likely reflecting genuine viewer feedback; the ground-truth is the top-liked comment, and distractors come from unrelated categories. In *Comment Popularity*, the model chooses the most-liked comment among four from the same video (top-1, top-10, top-50, top-100), based on subtle linguistic and contextual signals. All tasks follow a four-way single-choice format with one correct answer. The design emphasizes semantic confusion, distractor diversity, and option balance, ensuring that task success reflects genuine understanding rather than superficial cues or memorization. While some metadata-based tasks might appear straightforward, the overall suite is cognitively demanding. In particular, comment-related tasks (e.g., Comment Matching and Comment Popularity) require nuanced reasoning about social interaction, linguistic appeal, and collective preferences, making them non-trivial and discriminative.

The structured user-like thinking tasks in MIMIC-Bench are designed with varying levels of cognitive complexity. For clarity, we provide a more detailed discussion of task difficulty characteristics, along with representative challenging examples of the seven user-like thinking tasks, in Appendix D.

### 3.2.2 USER-LIKE MIMICKING TASK

Beyond structured reasoning, we introduce a novel task—**Comment Imitation**—to evaluate whether MLLMs can simulat human-like comments in social video contexts. This task emphasizes linguistic creativity, emotional expression, and social reasoning rather than factual correctness.

**Task Motivation.** As the second core component of MIMIC-Bench, this task captures the imaginative, affective, and context-aware nature of real viewer comments. It evaluates capability: generating authentic, human-style comments. Although the task inevitably involves subjective judgment, we explicitly design the protocol and aggregation strategy to ensure reproducibility and comparability, transforming subjective perception into a systematic evaluation dimension.

**Task Setup.** For each of the 4,000 benchmark videos, we collect the **top-5** most-liked real comments. Each of the **24 MLLMs** then generates one comment per video, producing a pool of 5 real and 24 model-generated comments, anonymized and randomly shuffled.

**Human Evaluation.** We conduct al human study using the same setup to provide a perceptual reference. These annotations serve as both a performance upper bound and an indicator of which AI-generated comments "deceive" human judges.

**Evaluation Metrics.** We use two metrics: **Imitation Quality**, based on the percentage of comments judged as human and their **average realism scores** (0–5). Beyond serving as a benchmark component, this protocol and its associated metrics also provide a methodological contribution, offering a reusable framework for future studies on human-likeness evaluation in multimodal generation.

This task creates a unique loop—*generation → judgment → scoring*—offering a lens to assess models' social and linguistic alignment with humans in multimodal settings.

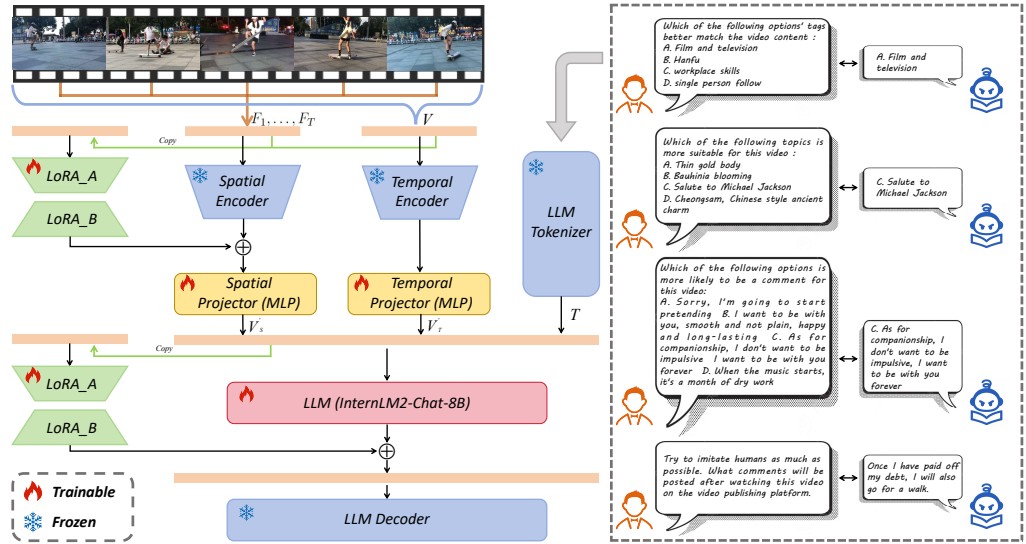

Figure 3: Overview of the proposed **MIMIC-Chat** architecture. Given sampled video frames, spatial and temporal features are extracted using dual encoders and projected into language space via lightweight MLPs. The projected visual tokens and task-specific instructions are fused and processed by a LoRA-enhanced InternLM2-Chat-8B model. The unified interface supports both structured reasoning tasks (*e.g.*, classification) and open-ended comment generation, enabling human-aligned video understanding.

## 4 MIMIC-CHAT

In this section, we present **MIMIC-Chat**, a unified multimodal framework tailored for human-aligned video understanding. The model supports both **User-like Thinking Tasks** and **User-like Mimicking Tasks** in **MIMIC-Bench**, enabling assessment of perception, cognition, and linguistic behavior within a cohesive pipeline.

### 4.1 ARCHITECTURE OVERVIEW

MIMIC-Chat is composed of three key components: a spatiotemporal video encoder, a task-guided instruction formatter, and a causal language model. This architecture supports both single-choice classification and free-form comment generation under a unified input format.

The model takes a video $V$ and task-specific instruction $T$ as input, and produces the output $Y$ via:

$$Y = \text{LM}([\texttt{VID}], \phi(V)', [\texttt{SEP}], T) \tag{1}$$

Here, $\phi(V)'$ denotes visual tokens extracted and projected from the video, while $[\texttt{VID}]$ and $[\texttt{SEP}]$ are special tokens marking the multimodal boundaries. This design enables multimodal fusion and instruction-following through a shared interface. It balances open-ended expressiveness with discriminative precision, making it suitable for all tasks in MIMIC-Bench. A detailed architecture diagram is shown in Figure 3.

### 4.2 VIDEO ENCODING AND VISUAL TOKEN EXTRACTION

To convert raw video into structured multimodal input, we adopt a spatiotemporal encoder with a lightweight projection layer, ensuring compatibility with the language model backbone.

**Frame Sampling and Preprocessing.** We adopt a dual-branch design: the spatial branch uniformly samples 8 frames to capture scene-level cues, while the temporal branch consumes the full frame sequence to preserve fine-grained dynamics and causal context. All frames are resized and center-cropped to a fixed resolution before encoding.

**Spatiotemporal Encoding via TimeSformer.** The processed frames are input to a spatiotemporal encoding pipeline composed of a spatial encoder (operating on sampled frames) and a temporal encoder (operating on the full sequence), which together capture complementary spatial layouts and temporal dependencies. It outputs a fixed-length token sequence:

$$\phi(V) = \{v_1, v_2, \ldots, v_N\}, \quad v_i \in \mathbb{R}^d \tag{2}$$

**MLP-Based Projection.** We align visual tokens with the language space via a lightweight MLP:

$$v'_i = \text{MLP}(v_i) \tag{3}$$

For simplicity, Eq. (3) shows a generic projection; in practice, we use distinct spatial and temporal projectors before fusion. The transformed sequence $\phi(V)'$ ensures effective vision-language integration. We distinguish a spatial projector (mapping tokens from the sampled-frame branch) and a temporal projector (mapping tokens from the full-sequence branch). Their outputs are subsequently fused within the language model via gated integration, ensuring complementary spatial–temporal reasoning.

**Multimodal Input Construction.** We concatenate the visual tokens $\phi(V)'$ with the natural language instruction $T$, using special tokens `[VID]` and `[SEP]` to delineate modalities. Each token is assigned modality-aware positional embeddings.

**Language Model Integration.** We employ **InternLM-8B** as the causal language model. It receives the multimodal sequence and generates either a structured classification output or a free-form textual response depending on the task type.

This video encoder module forms the foundation for MIMIC-Chat's high-level reasoning and generation capabilities, providing a consistent visual representation for diverse human-centered video understanding tasks. For a more detailed explanation of the model architecture and training configurations, refer to Appendix, where we provide additional insights on the system architecture and benchmark construction.

## 4.3 TRAINING DATA CONSTRUCTION - MIMIC-DATA

We build an instruction-tuning dataset named **Mimic-Data** from 150,000+ high-quality videos, each paired with multiple QA examples aligned with MIMIC-Bench tasks. This ensures consistency between training and evaluation in both structure and intent.

For the seven single-choice tasks (CIU, CAM, UIU), each sub-task is framed using a standardized prompt (*e.g.*, *"Based on the video, choose the most appropriate tag/title/comment"*), followed by four candidate options. The correct answer guides the model to align visual-linguistic cues with semantic labels. For the comment generation task, we use a unified prompt per video (*e.g.*, *"Generate a natural and imaginative comment for this video."*) and provide the top-5 most-liked user comments as references. This helps the model learn to express emotional nuance, associative thinking, and stylistic variation.

## 4.4 TRAINING OBJECTIVE AND OPTIMIZATION

We use supervised instruction tuning to train MIMIC-Chat on all tasks, optimizing for accurate, human-like responses to video and instructions.

**Training Loss.** All training samples are cast as question-answer pairs, allowing us to apply a standard language modeling loss:

$$\mathcal{L}_{\text{LM}} = -\sum_{t=1}^{|Y|} \log P(y_t \mid X, y_{<t}) \tag{4}$$

This formulation unifies structured classification (*e.g.*, choice "A") and open-ended generation (*e.g.*, full comments) under a single decoding objective.

**Multi-task Learning.** We employ a unified architecture without task-specific heads. Batches from different tasks are randomly sampled, enabling the model to generalize across diverse task types by interpreting the instruction and visual context.

**Optimization Strategy.** We fine-tune the full model except for the frozen visual backbone. Training leverages standard techniques including mixed-precision computation, dropout, gradient clipping, and label smoothing for improved stability and generalization. This unified objective aligns spatial–temporal semantics with human-centric reasoning goals, supporting both discriminative and generative tasks in MIMIC-Bench.

Table 1: Accuracy (%) on the seven reasoning tasks in MIMIC-Bench, covering three cognitive axes: Creator Intent Understanding (TiS: Title Selection, DeS: Description Selection), Content Attribute Matching (TaM: Tag Matching, ToM: Topic Matching, CaM: Category Matching), and User Interaction Understanding (CoM: Comment Matching, CoP: Comment Popularity). **Overall** denotes the average accuracy across all seven tasks. The highest, second-highest, and third-highest scores are highlighted in **purple**, **green**, and **pink**, respectively.

| Task Type | CIU | | CAM | | | UIU | | Overall↑ |
|---|---|---|---|---|---|---|---|---|
| **Models / Tasks** | TiS ↑ | DeS ↑ | TaM ↑ | ToM ↑ | CaM ↑ | CoM ↑ | CoP ↑ | |
| VideoChatGPT (Maaz et al., 2024) | 27.6 | 33.1 | 24.8 | 23.3 | 16.7 | 23.8 | 22.3 | 24.1 |
| Video-LLaVA (Lin et al., 2023) | 27.0 | 41.2 | 68.3 | 32.4 | 17.0 | 24.6 | 25.8 | 31.6 |
| VideoChat2 (Li et al., 2024b) | 37.7 | 33.2 | 30.1 | 46.1 | 34.2 | 32.9 | 26.7 | 33.6 |
| LLaVA-NeXT (Li et al., 2024a) | 32.7 | 32.9 | 50.4 | 35.3 | 31.4 | 25.7 | 23.5 | 31.6 |
| VideoLLaMA2 (Cheng et al., 2024) | 39.2 | 38.2 | 63.8 | 42.3 | 36.1 | 37.7 | 27.1 | 39.3 |
| VideoLLaMA3 (Zhang et al., 2025) | 78.7 | 58.0 | 78.6 | 82.0 | 49.5 | 51.3 | 28.3 | 58.7 |
| MiniCPM-V (Yao et al., 2024) | 77.6 | 65.4 | 77.0 | 78.6 | 49.1 | 55.0 | 28.9 | 59.2 |
| MiniCPM-o (Yao et al., 2024) | 71.1 | 60.7 | 84.8 | 83.3 | 50.2 | 54.2 | 32.0 | 59.5 |
| CogVLM2 (Hong et al., 2024) | 51.1 | 37.5 | 83.6 | 70.3 | 55.0 | 41.9 | 29.9 | 50.2 |
| Qwen2-VL (2B) (Wang et al., 2024a) | 48.6 | 51.9 | 70.8 | 57.1 | 47.3 | 36.2 | 24.3 | 44.3 |
| Qwen2-VL (7B) (Wang et al., 2024a) | 75.0 | 75.3 | 84.7 | 76.4 | 33.1 | 52.2 | 27.3 | 57.4 |
| Qwen2.5-VL (3B) (Bai et al., 2025) | 78.7 | 51.2 | 74.7 | 85.7 | 43.9 | 56.1 | 29.5 | 59.0 |
| Qwen2.5-VL (7B) (Wang et al., 2024a) | 80.8 | 54.1 | 72.6 | 88.0 | 43.6 | 58.1 | 29.0 | 59.9 |
| Qwen2.5-VL (72B) (Wang et al., 2024a) | 85.6 | 79.3 | 79.8 | 93.3 | 50.6 | 67.3 | 33.1 | 66.7 |
| InternVL2 (2B) (Chen et al., 2024b) | 46.1 | 27.2 | 67.3 | 62.7 | 45.1 | 33.5 | 23.8 | 41.9 |
| InternVL2 (4B) (Chen et al., 2024b) | 76.3 | 52.8 | 72.8 | 74.8 | 45.0 | 49.8 | 32.5 | 56.8 |
| InternVL2 (8B) (Chen et al., 2024b) | 84.1 | 72.9 | 78.6 | 87.2 | 51.1 | 63.9 | 30.6 | 64.4 |
| InternVL2.5 (4B) (Chen et al., 2024a) | 79.2 | 33.3 | 85.2 | 84.8 | 53.7 | 57.4 | 28.7 | 60.7 |
| InternVL2.5 (8B) (Chen et al., 2024a) | 83.5 | 47.6 | 86.6 | 89.8 | 50.0 | 64.0 | 31.6 | 64.5 |
| InternVideo2.5 (Wang et al., 2025b) | 83.2 | 71.7 | 87.5 | 90.1 | 53.5 | 64.4 | 32.7 | 66.3 |
| InternVL3 (78B) (Zhu et al., 2025) | 87.4 | 75.1 | 80.1 | 90.5 | 51.5 | 70.2 | 33.3 | 67.5 |
| ChatGPT-4o (Achiam et al., 2023) | 87.9 | 80.3 | 83.6 | 88.7 | 51.3 | 70.9 | 33.5 | 68.2 |
| Gemini2.5-pro (Comanici et al., 2025) | 92.6 | 89.5 | 82.9 | 92.3 | 56.1 | 82.9 | 43.5 | 75.1 |
| o3 (El-Kishky et al., 2025) | 93.2 | 86.1 | 85.7 | 92.1 | 55.2 | 77.4 | 45.5 | 74.6 |
| Human | 85.1 | 77.2 | 78.7 | 90.6 | 60.0 | 85.9 | 51.1 | 73.1 |
| **MIMIC-Chat(Ours)** | 90.4 | 87.1 | 86.7 | 92.5 | 55.7 | 78.3 | 43.6 | 74.1 |

**Parameter-Efficient Tuning.** Following (Hu et al., 2022), we integrate LoRA modules into select attention layers of InternLM-8B and jointly train them with the multimodal projector. This approach reduces memory usage while preserving performance. We further examine the contribution of the temporal encoder, spatial/temporal projectors, and LoRA through ablation studies in Appendix.

## 5 EXPERIMENTAL EVALUATION

### 5.1 EXPERIMENTAL SETUP

We evaluate MIMIC-Chat and 24 baselines—including 21 open-source MLLMs and 3 powerful proprietary models—as well as human participants on two key tracks in MIMIC-Bench: (1) **User-like Thinking Tasks**, which cover structured reasoning problems based on metadata; and (2) **User-like Mimicking Tasks**, which assess comment generation.

**Implementation Details.** For baseline models, we follow each model's official preprocessing strategy (including frame sampling and resolution). For MIMIC-Chat, we adopt the dual-branch design described in Section 4: the spatial branch uniformly samples 8 frames, while the temporal branch processes the full frame sequence. Prompts follow the format defined in Section 4. Subjective evaluation is conducted via a web interface. Three independent annotators judged each comment on two dimensions: (i) whether it was human- or AI-written, and (ii) a realism score between 0 and 5. We summarize key ablation findings here and provide detailed results in the Appendix. Removing the temporal encoder or either projector leads to substantial performance drops, while excluding LoRA causes consistent but smaller degradation.

Regarding inference efficiency, MIMIC-Chat requires approximately 24 GB GPU memory during evaluation and takes about 5–10 seconds to process a 20-second video clip on a single GPU.

### 5.2 USER-LIKE THINKING TASK PERFORMANCE

We evaluate all participants on the seven structured reasoning tasks in MIMIC-Bench, spanning three axes: creator intent understanding (CIU), content attribute matching (CAM), and user interaction understanding (UIU). Accuracy is reported per task and averaged across all subtasks. Results are summarized in Table 1.

Human participants achieve an overall accuracy of 73.1%, maintaining strong consistency across all reasoning axes and setting the upper bound on UIU tasks, particularly in Comment Matching (85.9%) and Comment Popularity (51.1%). **MIMIC-Chat**, our fine-tuned model, attains an overall accuracy of 74.1%, ranking third among all participants—slightly behind two frontier proprietary

models (Gemini-2.5-Pro at 75.1% and o3 at 74.6%) but surpassing all open-source baselines, including very large ones such as Qwen2.5-VL-72B (66.7%) and InternVL3-78B (67.5%). MIMIC-Chat achieves top-three performance across nearly all subtasks, with notable strength in creator intent understanding (TiS 90.4%, DeS 87.1%) and content attribute matching (CaM 55.7%).

Despite these gains, UIU remains the most challenging axis: while MIMIC-Chat improves substantially over prior open models, its performance on Comment Popularity (43.6%) still lags far behind human annotators. This highlights the intrinsic difficulty of modeling user-centric semantics and interaction cues.

A closer comparison across task types reveals that this gap is not primarily caused by perceptual limitations. Most models perform relatively better on perceptual or surface-aligned tasks (e.g., TiS and ToM), but struggle on tasks requiring higher-level social-cognitive reasoning (e.g., CaM and CoP), such as inferring human intent, emotion, and sociocultural cues beyond literal content. This pattern indicates that current failures stem more from insufficient human-centered commonsense reasoning than from visual understanding itself.

Overall, the results confirm that human-aligned fine-tuning enables MIMIC-Chat to close the gap with, and in some cases surpass, much larger models, underscoring the importance of task-specific alignment rather than sheer model scale alone.

## 5.3 User-like Mimicking Task Performance

We evaluate each model's ability to generate human-like comments through a comment simulation task. Table 2 summarizes the results. Here, the first column (*Models*) denotes the comment sources: each row reports evaluation results for comments generated by the corresponding model, while the *Human* row represents real user comments originally collected from video-sharing platforms. Additional details of the annotation protocol and bias mitigation strategies for the comment mimicking task are provided in Appendix D.5.

**Comment Simulation.**  For each video, all models generate one comment, which is then evaluated by human annotators. The simulation quality is measured in three ways: (1) the percentage of a model's comments judged as "human"; (2) the distribution of realism scores (0–5); and (3) the mean realism score per model. Each comment is anonymously presented to multiple human evaluators, who are asked to determine whether it was written by a human or an AI, and to rate its human-likeness on a 0–5 scale. Inter-annotator agreement reached 91.95%, calculated as the proportion of samples on which at least two annotators gave consistent judgments. This high level of agreement indicates that the evaluation is stable and less susceptible to individual annotator bias. The "Judged as Human" metric reflects the proportion of a model's comments that were classified as human-written. The "Score@k" distribution indicates the percentage of all comments receiving score $k$, and "Mean Score" reports the overall average realism score across the model's 4,000 generated comments.

As shown in the table, **MIMIC-Chat significantly outperforms all baseline models** in the key metric of being judged as human (64.24%), more than triple most existing models, and second only to real human comments (87.57%). Its average realism score (2.88) is also notably higher than that of other models. These results suggest that MIMIC-Chat produces more natural, human-aligned comments in both linguistic style and semantic coherence. These findings highlight that MIMIC-Chat achieves the most human-aligned comment generation performance among all evaluated models.

We attribute this advantage to a fundamental behavioral difference: most baseline models tend to generate literal or content-descriptive comments—such as rephrasing video scenes or restating visual facts—which are easily recognized as AI-written by human annotators. In contrast, MIMIC-Chat demonstrates a stronger capacity for simulating human-like thinking patterns, often exhibiting signs of *divergent, associative, or emotionally reflective reasoning*—traits more characteristic of genuine human responses in open-domain video discussions. Beyond open-source baselines, the inclusion of stronger proprietary models (e.g., ChatGPT-4o, Gemini2.5-pro, and o3) reveals a narrowing gap in human-likeness. These models achieve substantially higher realism scores and human-judgment rates than most open-source MLLMs. Nevertheless, MIMIC-Chat remains the most competitive among open-source systems and still exhibits a notable margin over proprietary counterparts in the "Judged as Human" metric. Taken together, these results indicate that while recent closed-source models are improving rapidly in simulating human-like comments, there remains

Table 2: Evaluation of comment simulation across all participants, based on human judgments. The results include human-likeness classification rates, realism score distributions (0–5), and mean scores. The highest, second-highest, and third-highest scores are highlighted in **purple**, **green**, and **pink**, respectively.

| Models\Task | Judged as Human (%) ↑ | Judged as AI (%) ↓ | Score Distribution(%) | | | | | | Mean Score ↑ |
|---|---|---|---|---|---|---|---|---|---|
| | | | Score@0 ↓ | Score@1 ↓ | Score@2 | Score@3 | Score@4 ↑ | Score@5 ↑ | |
| VideoChatGPT (Maaz et al., 2024) | 18.65 | 81.35 | 50.15 | 27.52 | 3.67 | 7.95 | 6.27 | 4.43 | 1.06 |
| Video-LLaVA (Lin et al., 2023) | 6.30 | 93.70 | 63.16 | 28.48 | 1.88 | 2.07 | 2.44 | 1.97 | 0.58 |
| VideoChat2 (Li et al., 2024b) | 21.02 | 78.98 | 53.66 | 21.54 | 3.79 | 9.40 | 7.70 | 3.92 | 1.08 |
| LLaVA-NeXT (Li et al., 2024a) | 7.02 | 92.98 | 63.20 | 27.58 | 2.12 | 2.93 | 1.90 | 2.27 | 0.60 |
| VideoLLaMA2 (Cheng et al., 2024) | 2.70 | 97.30 | 75.57 | 20.37 | 1.25 | 0.31 | 1.56 | 0.94 | 0.35 |
| VideoLLaMA3 (Zhang et al., 2025) | 4.02 | 95.98 | 68.46 | 24.95 | 2.47 | 1.28 | 1.19 | 1.65 | 0.47 |
| MiniCPM-V (Yao et al., 2024) | 8.17 | 91.83 | 61.78 | 26.98 | 2.99 | 2.64 | 2.37 | 3.25 | 0.67 |
| MiniCPM-o (Yao et al., 2024) | 6.85 | 93.15 | 66.43 | 24.60 | 2.11 | 1.23 | 2.81 | 2.81 | 0.58 |
| CogVLM2 (Hong et al., 2024) | 7.50 | 92.50 | 68.20 | 21.40 | 2.80 | 4.80 | 1.70 | 1.10 | 0.54 |
| Qwen2-VL (2B) (Wang et al., 2024a) | 6.54 | 93.46 | 63.82 | 27.25 | 2.39 | 2.08 | 2.31 | 2.16 | 0.58 |
| Qwen2-VL (7B) (Wang et al., 2024a) | 6.99 | 93.01 | 62.36 | 28.25 | 2.33 | 2.70 | 1.95 | 2.40 | 0.61 |
| Qwen2.5-VL (3B) (Bai et al., 2025) | 14.18 | 85.82 | 59.61 | 23.76 | 2.38 | 3.93 | 4.83 | 5.49 | 0.87 |
| Qwen2.5-VL (7B) (Bai et al., 2025) | 10.25 | 89.75 | 67.76 | 20.73 | 1.19 | 2.60 | 3.27 | 4.46 | 0.66 |
| InternVL2 (2B) (Chen et al., 2024b) | 1.03 | 98.97 | 80.59 | 17.66 | 0.66 | 0.29 | 0.29 | 0.51 | 0.24 |
| InternVL2 (4B) (Chen et al., 2024b) | 3.69 | 96.31 | 77.14 | 17.70 | 1.47 | 0.66 | 1.55 | 1.47 | 0.36 |
| InternVL2 (8B) (Chen et al., 2024b) | 5.37 | 94.63 | 66.09 | 26.36 | 2.11 | 2.18 | 1.09 | 2.18 | 0.52 |
| InternVL2.5 (4B) (Chen et al., 2024a) | 16.26 | 83.74 | 54.24 | 26.58 | 2.93 | 4.39 | 5.70 | 6.16 | 0.99 |
| InternVL2.5 (8B) (Chen et al., 2024a) | 10.90 | 89.10 | 59.47 | 26.54 | 2.86 | 2.56 | 4.21 | 4.36 | 0.79 |
| InternVideo2.5 (Wang et al., 2025b) | 12.67 | 87.33 | 55.58 | 27.60 | 3.77 | 4.45 | 4.45 | 4.15 | 0.87 |
| Qwen2.5-VL (72B) (Bai et al., 2025) | 26.93 | 73.07 | 42.59 | 22.83 | 7.65 | 6.92 | 9.63 | 10.40 | 1.49 |
| InternVL3 (78B) (Zhu et al., 2025) | 30.21 | 69.79 | 41.23 | 18.87 | 9.69 | 5.48 | 13.83 | 10.90 | 1.65 |
| ChatGPT-4o (Achiam et al., 2023) | 40.32 | 59.68 | **28.65** | 21.32 | 9.71 | 10.91 | 17.64 | 11.77 | 2.03 |
| Gemini2.5-pro (Comanici et al., 2025) | 41.87 | 58.13 | 29.51 | **16.84** | 11.78 | 9.97 | 18.31 | **13.59** | **2.12** |
| o3 (El-Kishky et al., 2025) | **43.36** | **56.64** | 28.68 | 19.71 | 8.25 | 11.33 | **19.56** | 12.47 | 2.11 |
| Human | **87.57** | **12.43** | **8.84** | **3.13** | 0.49 | 4.37 | **24.59** | **58.58** | **4.08** |
| **MIMIC-Chat (ours)** | **64.24** | **35.76** | **7.62** | **15.23** | 12.91 | 26.82 | **20.20** | **17.22** | **2.88** |

a sizable gap compared to genuine human responses, underscoring the challenge of fully capturing the subtleties of human reasoning and expression.

To ensure fairness, we also fine-tuned several strong open baselines on MIMIC-Data under identical settings. While their performance improved, they still lagged behind MIMIC-Chat, suggesting that our advantage stems not merely from access to task-aligned data but from the proposed architecture and training strategy. Full results are reported in Appendix.

In addition, to examine potential biases introduced by engagement-based video selection and to assess robustness beyond the benchmark distribution, we conduct additional out-of-domain and cross-benchmark evaluations. Detailed analyses and results are reported in Appendix E.

# 6 CONCLUSION

We presented **MIMIC-Data** and **MIMIC-Bench**, a large-scale dataset and benchmark for evaluating human-aligned video understanding, and introduced **MIMIC-Chat**, a model trained to bridge the gap between perception and human-like reasoning. Across 24 competitive MLLMs, results show that while recent systems advance perception, they still fall short in mimicking human thought and expression. MIMIC-Chat narrows this gap, setting a new state of the art among open-source models. We believe this work opens a new direction toward socially and cognitively aligned multimodal intelligence.

# ACKNOWLEDGEMENT

This work was supported in part by the National Natural Science Foundation of China under Grants 62401365, 62225112, 62271312, 62132006, U24A20220, and in part by the China Postdoctoral Science Foundation under Grants BX20250411, 2025M773473.

ETHICS STATEMENT

Our work adheres to the ICLR Code of Ethics. This work introduces MIMIC-Bench, a benchmark designed to evaluate multimodal large language models (MLLMs) on human-aligned video understanding tasks, and MIMIC-Chat, a model fine-tuned for this purpose.

All videos and metadata in MIMIC-Data were collected exclusively from publicly accessible sources. We only include content that is publicly available at the time of collection and do not access private, restricted, or paywalled material. We respect the copyright policies and terms of service of the hosting platforms. The benchmark does not redistribute raw video files; instead, we release annotations, benchmark splits, and associated metadata necessary for research evaluation. Users are required to retrieve original content directly from the respective hosting platforms under their applicable terms of service. We also provide a contact channel to accommodate content removal requests if needed.

To protect privacy, we exclude personally identifiable information (PII) during data collection and curation. We do not release uploader identities, user handles, account IDs, contact information, or other identifying metadata. Content involving private individuals, minors, sensitive personal circumstances, or identifiable private locations is excluded based on predefined filtering criteria. Both automated filtering rules and manual review procedures are applied to minimize the inclusion of sensitive or personal content.

To mitigate risks of reputational harm, we exclude videos or annotations that contain defamatory statements, targeted harassment, unverified allegations, or highly sensitive personal topics (e.g., medical conditions, criminal accusations, or political attacks directed at specific individuals). The dataset is intended strictly for academic research and benchmarking purposes. It is not designed for profiling, surveillance, or other applications that could adversely impact individuals or communities.

Further details on dataset collection procedures, copyright compliance measures, de-identification rules, exclusion criteria, and responsible release protocols are provided in the Appendix.

REPRODUCIBILITY STATEMENT

We have made significant efforts to ensure the reproducibility of our work. The main paper details the construction process of MIMIC-Bench, the task definitions, and the evaluation protocols. We describe the design of MIMIC-Chat, including its dual-branch spatial–temporal encoder, training strategy, and optimization settings. Appendix C and D provide further details on model architecture, training configurations, dataset construction, and visualization examples. We plan to release the benchmark data splits, evaluation code, and MIMIC-Chat training scripts upon publication to facilitate replication and follow-up research.

LLM USAGE STATEMENT

Large Language Models (LLMs) were used only as part of the evaluation in our benchmark, where 24 baseline models (both open-source and proprietary) were compared against our proposed MIMIC-Chat. LLMs were not used to generate any dataset samples, annotations, or experimental results. All dataset construction, annotation, and analysis were conducted independently by the authors. We used LLMs in a limited capacity for language polishing of the manuscript, but no section of the research design, data collection, or core analysis was generated by LLMs. The authors take full responsibility for the content of this paper.

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

# A ABLATION STUDIES

## A.1 ABLATION SETUP

To isolate the contribution of each key component in **MIMIC-Chat**, we perform controlled ablations under a unified training and evaluation pipeline. Unless otherwise specified, all settings strictly follow the main paper (§4–§5): same training data (**MIMIC-Data**), instruction format, loss, optimizer, batch size, number of epochs, inference hardware (A6000 GPUs), and evaluation metrics. For thinking tasks we report accuracy on the seven multiple-choice subtasks; for mimicking we report human-judged comment simulation metrics (*Judged as Human*, Score@$k$, Mean Score). We do not include source-identification results in the appendix.

**Backbone and Inputs (constant across variants).** We use the full **dual-branch** video pathway described in §4 as the reference ("Full"): a *spatial branch* that uniformly samples 8 frames for scene-level cues, and a *temporal branch* that processes the full frame sequence for fine-grained dynamics. Visual tokens from both branches are projected by branch-specific MLP projectors and fused into the language model with gated integration. The causal LM is **InternLM-8B** with LoRA modules enabled in the Full configuration.

**Ablated Variants.** We construct four ablation variants by toggling one component at a time while keeping all other factors identical:

- **w/o LoRA** (*No-LoRA*): Disable all LoRA adapters in InternLM-8B and freeze the LM parameters; only the multimodal projector(s) remain trainable. This tests the role of parameter-efficient language adaptation for instruction alignment.

- **w/o Temporal Encoder** (*No-TempEnc*): Remove the temporal branch (full-sequence processing); the model uses only the spatial branch with 8 uniformly sampled frames. This probes the importance of explicit temporal modeling.

- **w/o Temporal Projector** (*No-TempProj*): Keep both branches but remove the temporal-specific projector; temporal tokens are routed through the spatial projector (shared MLP) before fusion. This examines whether a dedicated temporal projection space is necessary.

- **w/o Spatial Projector** (*No-SpatProj*): Keep both branches but remove the spatial-specific projector; spatial tokens are routed through the temporal projector (shared MLP). This complements the previous variant and tests sensitivity to projector specialization.

**Fairness Controls.** All variants use the same prompts and decoding settings as the Full model. Frame sampling, resolution, and preprocessing follow §5. When a branch is removed (e.g., *No-TempEnc*), the remaining branch and its projector are unchanged; when a projector is removed (e.g., *No-TempProj/No-SpatProj*), tokens are passed through the remaining projector to keep token dimensionality and downstream interfaces intact. This design ensures that any performance difference can be attributed to the ablated component rather than confounds in optimization or data.

## A.2 QUANTITATIVE COMPARISON ON THINKING TASKS

We quantify the contribution of each core component on the seven multi-choice tasks (CIU: TiS/DeS; CAM: TaM/ToM/CaM; UIU: CoM/CoP). All settings (data, preprocessing, prompts) are held fixed as in the main experiments; only the indicated module is removed or altered.

**Key observations.** The ablation results reveal that both temporal modeling and LoRA-based adaptation are indispensable. Removing the *temporal encoder* causes the largest overall drop (74.1 → 65.8; −8.3 pp), with especially severe declines in *DeS* (−33.8 pp) and UIU tasks (CoM: −12.9 pp, CoP: −11.0 pp), underscoring the need for end-to-end temporal reasoning. LoRA adaptation is equally critical: ablating it reduces Overall to 66.9 (−7.2 pp), driven by sharp declines on semantics-heavy tasks such as *DeS* (−22.7 pp), *TaM* (−16.3 pp), and both UIU subtasks. The *temporal projector* further contributes complementary gains, as its removal lowers Overall to 67.5 (−6.6 pp) and disproportionately weakens CAM and UIU performance, while the *spatial projector* supports static semantics, with its absence (Overall 68.7; −5.4 pp) most affecting *DeS* (−18.9 pp). Together,

Table 3: Accuracy (%) on the seven structured reasoning tasks after ablating components of MIMIC-Chat. CIU includes Title Selection (TiS) and Description Selection (DeS); CAM includes Tag/Topic/Category Matching (TaM/ToM/CaM); UIU includes Comment Matching (CoM) and Comment Popularity (CoP). **Overall** is the average over all seven tasks. The full model's scores are highlighted in purple.

| Task Type | CIU | | CAM | | | UIU | | Overall↑ |
|---|---|---|---|---|---|---|---|---|
| Models / Tasks | TiS ↑ | DeS ↑ | TaM ↑ | ToM ↑ | CaM ↑ | CoM ↑ | CoP ↑ | |
| **MIMIC-Chat (full)** | **90.4** | **87.1** | **86.7** | **92.5** | **55.7** | **78.3** | **43.6** | **74.1** |
| *w/o LoRA* | 88.3 | 64.4 | 70.4 | 89.8 | 47.3 | 68.2 | 34.1 | 66.9 |
| *w/o Temporal Encoder* | 85.6 | 53.3 | 87.5 | 89.5 | 51.4 | 65.4 | 32.6 | 65.8 |
| *w/o Temporal Projector* | 89.0 | 75.4 | 72.8 | 91.8 | 48.6 | 70.3 | 34.5 | 67.5 |
| *w/o Spatial Projector* | 89.3 | 68.2 | 84.2 | 91.7 | 50.2 | 72.6 | 35.1 | 68.7 |

these findings highlight that temporal components drive interaction- and context-sensitive reasoning, while LoRA secures creator-intent and textual alignment, and only their integration allows the full model to achieve a balanced advantage across CIU, CAM, and UIU.

### A.3 DISCUSSION

The ablation results highlight several broader implications. First, **temporal modeling and LoRA-based language adaptation are complementary**: the former underpins interaction- and context-sensitive reasoning (UIU, CAM), while the latter ensures fine-grained textual alignment (CIU, DeS/TaM). Second, different task categories stress distinct modalities—CIU benefits most from semantic adaptation, whereas UIU requires strong temporal grounding—indicating that balanced multimodal integration is essential for generalizable video understanding. Finally, the consistent superiority of the full model suggests that parameter-efficient language tuning and temporally-aware encoding are not just additive improvements, but jointly critical for bridging the gap between perception-driven reasoning and socially aligned interpretation, reinforcing the design principles behind MIMIC-Chat.

## B FINE-TUNING OTHER MODELS

### B.1 SETUP

To assess whether the improvements of MIMIC-Chat stem solely from access to MIMIC-Data, we fine-tuned several strong video-language models under identical conditions. Specifically, we selected three representative backbones covering different architectures and scales: **Qwen2.5-VL-7B**, **InternVL2.5-8B**, and **InternVideo2.5-8B**. These models were chosen because of their strong baseline performance and wide adoption in the community.

For fairness, all models were fine-tuned on the same training split of **MIMIC-Data** that was used to train MIMIC-Chat, and evaluated on the official test set of **MIMIC-Bench**. The fine-tuning procedure followed a unified setup:

- **Data.** The full MIMIC-Data training split was used without task-specific resampling. All structured and generative tasks share the same preprocessing pipeline.

- **Optimization.** We employed AdamW optimizer with a learning rate of $2 \times 10^{-5}$, cosine decay, and batch size of 128. Training was run for 3 epochs with early stopping on the validation set.

- **LoRA.** For parameter-efficient adaptation, LoRA modules were applied to the language backbone of each model, while vision encoders were kept frozen. This ensured efficiency and comparability across different backbones.

- **Evaluation.** All models were evaluated on the seven structured reasoning tasks (CIU, CAM, UIU) and the mimicking tasks, using accuracy for multi-choice and human-likeness metrics for generative tasks.

Table 4: Accuracy (%) on the seven structured reasoning tasks after fine-tuning existing models on MIMIC-Data. CIU includes Title Selection (TiS) and Description Selection (DeS); CAM includes Tag/Topic/Category Matching (TaM/ToM/CaM); UIU includes Comment Matching (CoM) and Comment Popularity (CoP). **Overall** is the average over all seven tasks. The full model's scores are highlighted in  purple .

| Task Type | CIU | | CAM | | | UIU | | Overall↑ |
| Models / Tasks | TiS ↑ | DeS ↑ | TaM ↑ | ToM ↑ | CaM ↑ | CoM ↑ | CoP ↑ | |
| --- | --- | --- | --- | --- | --- | --- | --- | --- |
| Qwen2.5-VL-7B | 80.8 | 54.1 | 72.6 | 88.0 | 43.6 | 58.1 | 29.0 | 59.9 |
| InternVL2.5-8B | 83.5 | 47.6 | 86.6 | 89.8 | 50.0 | 64.0 | 31.6 | 64.5 |
| InternVideo2.5-8B | 83.2 | 71.7 | 87.5 | 90.1 | 53.5 | 64.4 | 32.7 | 66.3 |
| *fine-tuned on MIMIC-Data* | | | | | | | | |
| Qwen2.5-VL-7B (ft) | 85.1 | 60.2 | 80.1 | 89.7 | 46.8 | 64.7 | 32.3 | 66.4 |
| InternVL2.5-8B (ft) | 85.6 | 53.3 | 87.5 | 89.5 | 51.4 | 65.4 | 32.6 | 65.8 |
| InternVideo2.5-8B (ft) | 87.5 | 76.2 | 90.3 | 91.3 | 51.3 | 66.9 | 33.1 | 68.1 |
| **MIMIC-Chat (Ours)** | **90.4** | **87.1** | **86.7** | **92.5** | **55.7** | **74.3** | **43.6** | **74.1** |

This setup ensures that any observed performance differences are attributable to model architecture and adaptation capacity, rather than data imbalance or training procedure.

## B.2 RESULTS

We report the performance of fine-tuned models compared with MIMIC-Chat across both structured reasoning and mimicking tasks.

**Key findings.** Fine-tuning on MIMIC-Data consistently improves all baseline models, with Qwen2.5-VL-7B gaining from 59.9 to 66.4 overall accuracy and InternVideo2.5-8B rising from 66.3 to 68.1. The strongest gains appear in semantics-heavy tasks such as DeS (e.g., +6.1 for Qwen2.5-VL-7B) and TaM (+7.5), highlighting the value of domain-specific alignment. Nevertheless, MIMIC-Chat remains clearly ahead across nearly all tasks, especially in DeS (87.1 vs. 76.2 for the best fine-tuned baseline) and CoP (43.6 vs. 33.6), confirming that its superior architecture and training design contribute substantially beyond data fine-tuning alone.

## B.3 DISCUSSION

The fine-tuning experiments demonstrate that existing MLLMs, when adapted to MIMIC-Data, can indeed improve their performance on user-centric reasoning tasks. Models such as Qwen2.5-VL-7B and InternVideo2.5-8B show consistent gains in both creator-intent understanding (e.g., DeS) and content-attribute matching (e.g., TaM), validating the importance of training data that reflects human communicative patterns. However, the improvements are incremental: even the best fine-tuned baselines remain substantially behind MIMIC-Chat in both overall accuracy and in the most challenging sub-tasks. This suggests that while domain-specific fine-tuning enhances alignment, it cannot substitute for architectural innovations and multi-stage training pipelines explicitly designed for human-like reasoning. In other words, access to the same data is not sufficient—MIMIC-Chat's advantage lies in how it integrates LoRA-based language adaptation, temporal-spatial modeling, and task-specific objectives to achieve balanced and robust performance across all axes of MIMIC-Bench.

## C MODEL ARCHITECTURE AND IMPLEMENTATION DETAILS

In Section 4 of the main paper, we provided a brief overview of the MIMIC-Chat architecture, which integrates a video encoder, an instruction formatter, and a language model into a unified framework. This section supplements that overview by elaborating on implementation-level details and training configurations, including hardware setup, fine-tuning strategies, visual input processing, and optimization techniques.

## C.1 TRAINING ENVIRONMENT AND HARDWARE CONFIGURATION

All experiments were conducted on a high-performance server equipped with six NVIDIA RTX A6000 GPUs (each with 48 GB memory), using CUDA 12.2 and driver version 535.179. Model training was implemented with PyTorch, and distributed optimization was realized through `torchrun` and DeepSpeed Stage 1, enabling parameter offloading and mixed-precision (bf16) training for enhanced memory and efficiency.

## C.2 FINE-TUNING STRATEGY AND MODULE CONFIGURATION

MIMIC-Chat adopts a parameter-efficient instruction tuning strategy, updating only key components:

- **Language Model (LLM)**: LoRA modules are injected into the attention sublayers of InternLM2-Chat-8B, leveraging low-rank adaptation to reduce trainable parameters.
- **Freezing Strategy**: Both the vision backbone and LLM backbone are frozen during training, while projection layers and LoRA modules remain trainable.
- **Vision Projection**: Spatial and temporal features are independently projected into the language space via two MLPs to preserve visual-linguistic alignment.

Additionally, bf16 mixed-precision training and gradient checkpointing are enabled to reduce memory usage without compromising performance.

## C.3 VIDEO INPUT PROCESSING AND VISUAL TOKEN CONSTRUCTION

Each video sample undergoes standardized frame sampling and preprocessing:

- **Frame Sampling**: The spatial encoder uses 8 frames uniformly sampled from each video, while the temporal encoder processes the full sequence of frames to capture fine-grained temporal dynamics.
- **Image Processing**: All frames are center-cropped and resized to 448×448 resolution.
- **Feature Encoding**: The spatial and temporal encoders extract static and dynamic information. The spatial encoder processes the 8 uniformly sampled frames, while the temporal encoder consumes the full frame sequence to capture temporal continuity.
- **Projection and Tokenization**: Features from both spatial and temporal encoders are projected into the language model token space to form visual tokens.
- **Input Construction**: Visual tokens and natural language instructions are concatenated, with [VID] and [SEP] tokens denoting modality boundaries.

A dynamic patch control mechanism (up to 6 patches) and thumbnail token injection are introduced to accommodate longer videos and enhance contextual representation.

## C.4 TRAINING CONFIGURATION AND OPTIMIZATION

To ensure performance and stability, we adopt the following training settings:

- Epochs: 50
- Per-device batch size: 2; Global batch size: 4 (via gradient accumulation)
- Learning rate: 4e-5 with 3% warm-up
- Input resolution: 448×448
- Max dynamic patches: 6
- Optimizer: AdamW with weight decay 0.01
- Scheduler: Cosine decay
- Gradient clipping: enabled

- Max sequence length: 4096 tokens
- Grouped training: samples are grouped by token length to accelerate convergence
- Monitoring: training logs recorded via TensorBoard; best-performing checkpoints and LoRA weights are saved periodically

### C.5 ENGINEERING OPTIMIZATIONS FOR SYSTEM ROBUSTNESS

To support long-context, large-scale multimodal training, we introduce several engineering enhancements:

- **Lazy-loading dataset class** for robust video streaming with corrupted frame handling;
- **Custom trainer** with LoRA-only weight saving to facilitate model deployment and ablation analysis;
- **Dynamic image preprocessing** that adapts patch numbers and resolutions on-the-fly to control memory usage;
- **Multi-task training support**, enabling unified classification and generation under instruction-based prompts.

## D BENCHMARK CONSTRUCTION AND VISUALIZATION EXAMPLES

In Section 3 of the main paper, we outlined the construction of MIMIC-Bench and the motivation for its design. This section provides additional implementation details regarding how the 4,000 benchmark videos were selected and scored, including the criteria used to ensure their human-centric relevance and linguistic richness. It also supplements the dataset composition and preparation steps that underpin our evaluation tasks.

### D.1 SELECTION AND SCORING CRITERIA

To ensure that the benchmark accurately reflects human-style interpretation and communicative behavior, we curated 4,000 videos from the larger MIMIC-Data pool of 150,000+ user-shared videos. The selection process involved a multi-stage filtering pipeline:

**(1) Engagement Scoring.** Each video was assigned a composite engagement score to measure real-world user interaction. The score combines the log-normalized values of like count, favorite count, share count, and comment count, computed as:

$$\text{Engagement Score} = \alpha \cdot \log(Like) + \beta \cdot \log(Favorite) + \gamma \cdot \log(Share) + \delta \cdot \log(Comment) \quad (5)$$

We set $\alpha = 1.0$, $\beta = 0.8$, $\gamma = 0.5$, and $\delta = 1.2$ to place greater emphasis on comments, which better reflect human intent and understanding.

**(2) Metadata Integrity.** After sorting by engagement score, we retained only those videos with complete metadata fields, including title, description, tags, and topic. We further ensured that each video contains at least five unique, high-quality user comments and is free from decoding errors or anomalously short durations.

**(3) Semantic Coverage and Diversity.** To ensure diverse coverage across topics and expression styles, we adopted the following constraints:

- Top 2% videos from TikTok and top 5% from YouTube were selected.
- The selected pool spans 8 major categories (e.g., lifestyle, travel, beauty) and 20+ subcategories.
- The distribution of comment types was controlled to include exclamatory, inquisitive, associative, and ironic styles.

This multi-dimensional curation strategy ensures that MIMIC-Bench captures both high user engagement and rich human-centered semantics, laying a robust foundation for downstream evaluation of multimodal models on user-aligned reasoning and mimicking capabilities.

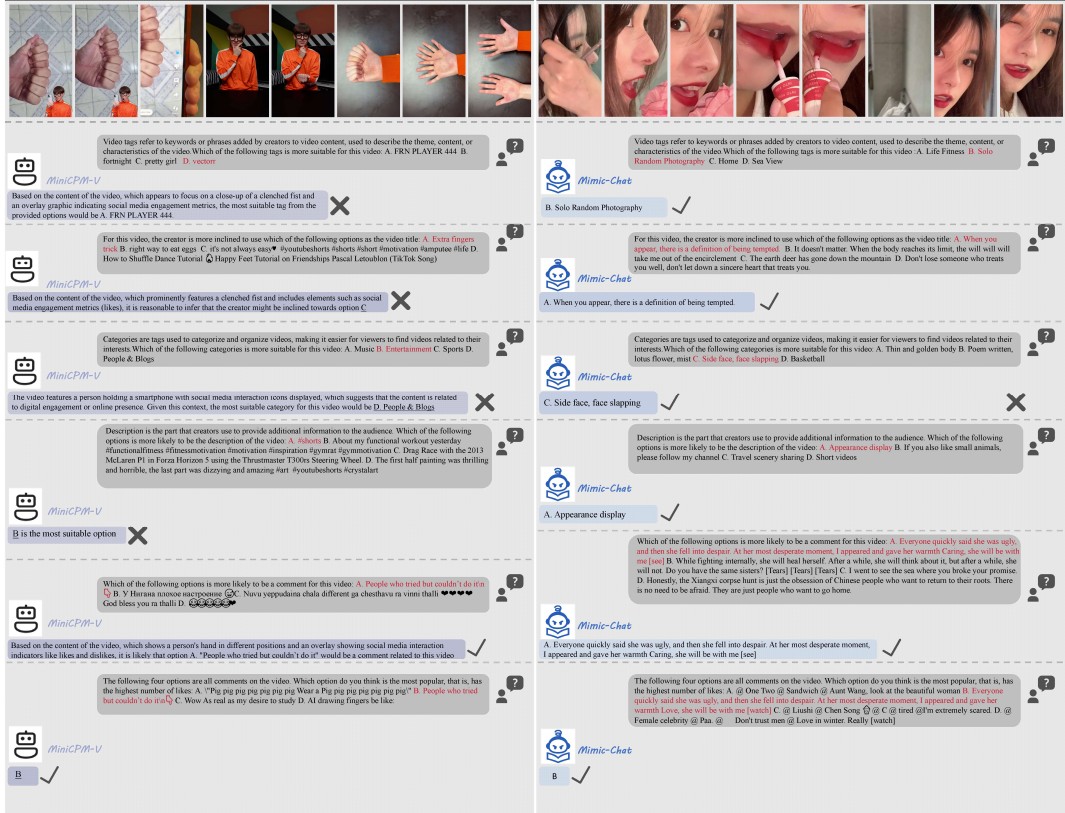

Figure 4: Illustration of the responses from the **MiniCPM-V** and **MIMIC-Chat (Ours)** models on user-like reasoning tasks. Ground truth is marked in red in the question, and model responses and correctness follow the corresponding question.

## D.2 QUALITATIVE EXAMPLES OF MODEL RESPONSES

To better illustrate the performance of different multimodal large language models (MLLMs) on MIMIC-Bench tasks, we present a series of representative model response examples in this supplementary material. These examples cover a variety of tasks such as title selection, tag matching, comment imitation, demonstrating each model's ability to interpret real-world user videos and generate human-aligned responses.

Each example includes the following components:

- **Task input**: the multimodal metadata associated with the video, along with the task prompt;

- **Model-generated responses**: the outputs from a set of baseline MLLMs, as well as our proposed MIMIC-Chat model;

- **Ground-truth or reference answers**: provided for comparison to evaluate model correctness or human-likeness.

As shown in Figures 4, 5, and 6, the visualized outputs display the input prompts, the model predictions, and whether the generated results match the expected answers. These examples qualitatively complement the quantitative results in the main paper, highlighting each model's strengths and weaknesses across tasks involving higher-level reasoning, creative intent recognition, and user interaction interpretation.

We hope these examples will deepen understanding of the challenges posed by MIMIC-Bench and inspire the development of more human-aligned multimodal systems.

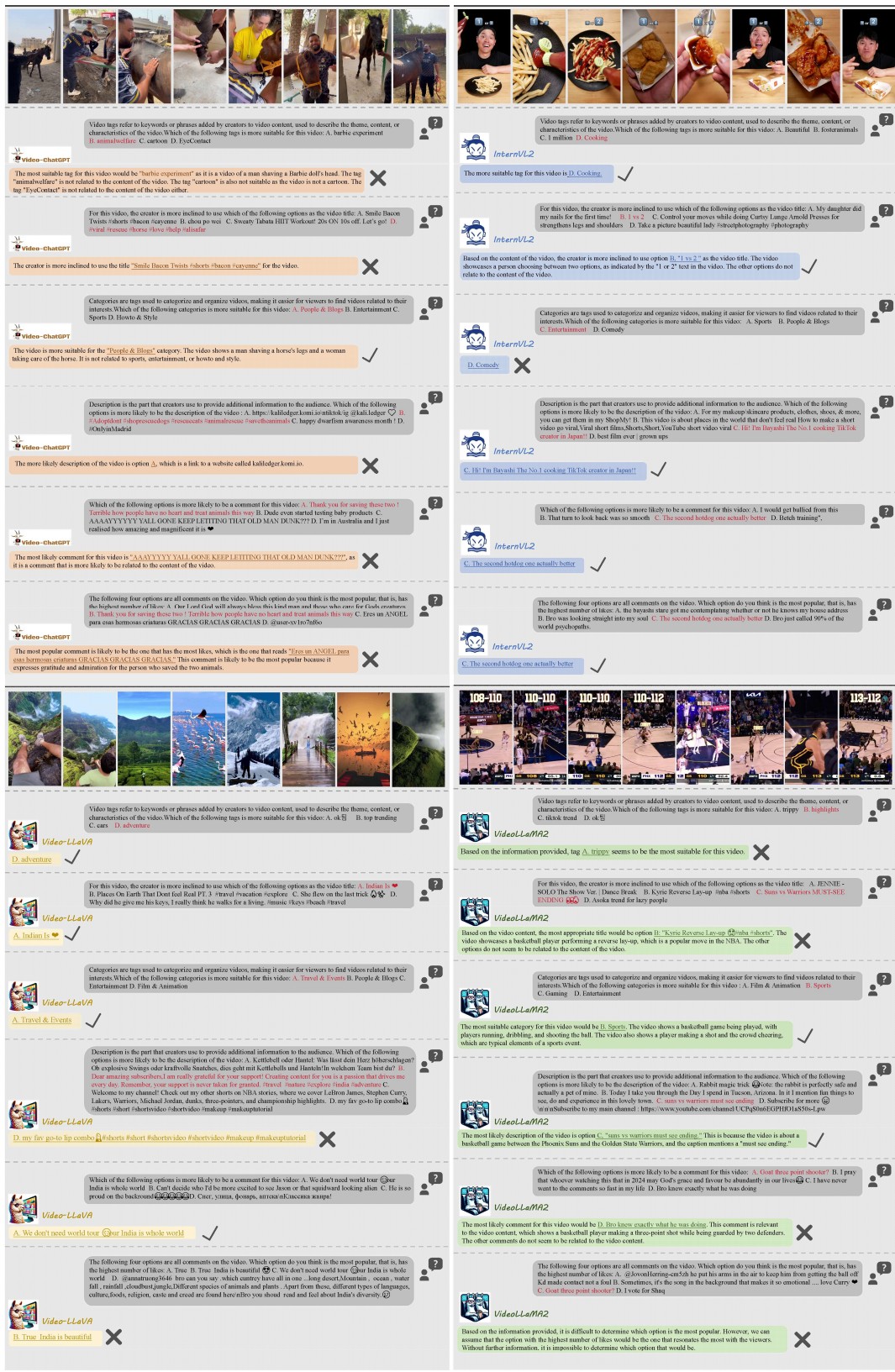

Figure 5: Illustration of the responses from the **Video-ChatGPT**, **InternVL2**, **Video-LLaVA**, and **VideoLLaMA2** models on user-like reasoning tasks. Ground truth is marked in red in the question, and model responses and correctness follow the corresponding question.

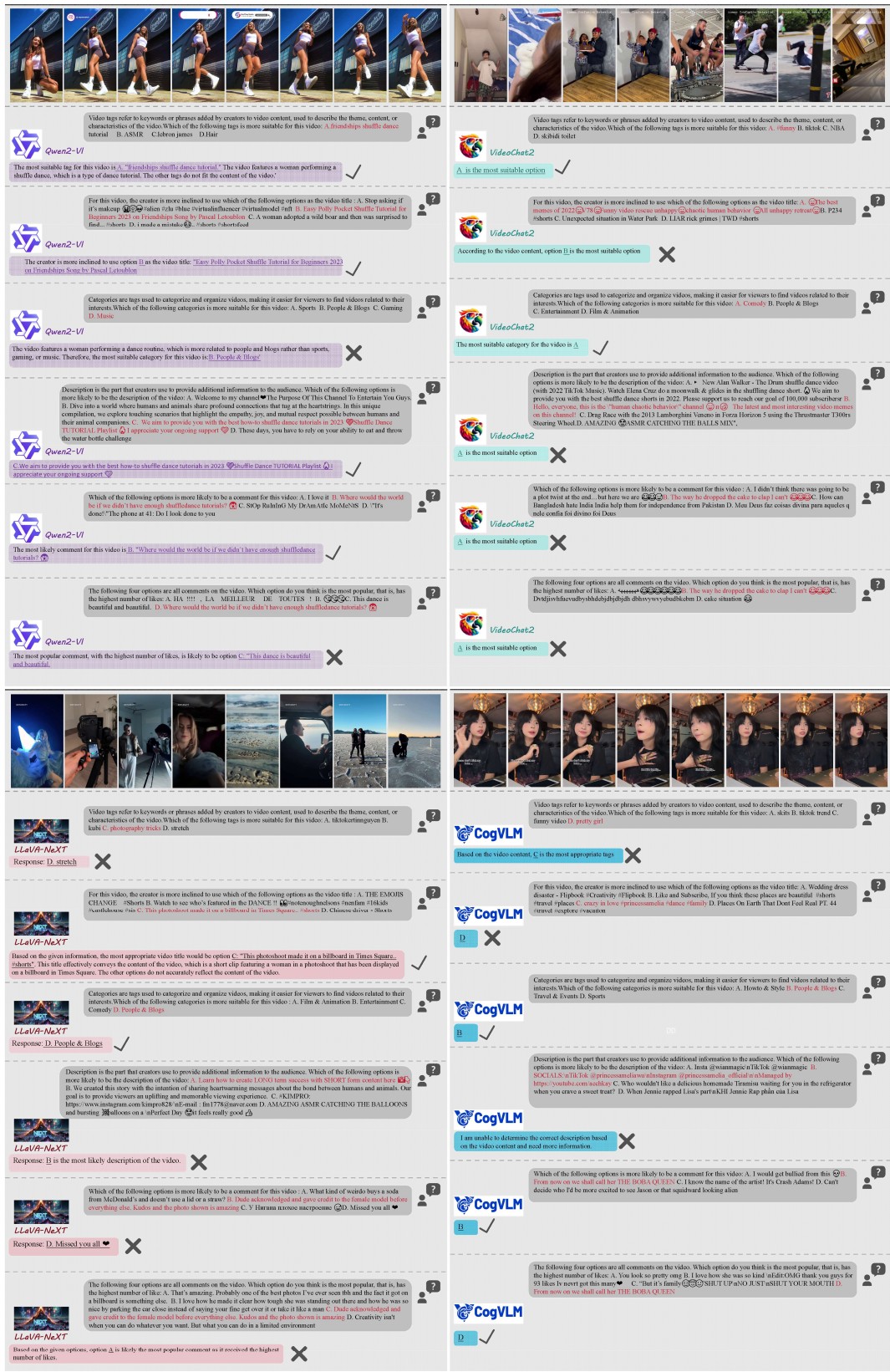

Figure 6: Illustration of the responses from the **Qwen2-VL**, **VideoChat2**, **LLaVA-NeXT**, and **CogVLM2** models on user-like reasoning tasks. Ground truth is marked in red in the question, and model responses and correctness follow the corresponding question.

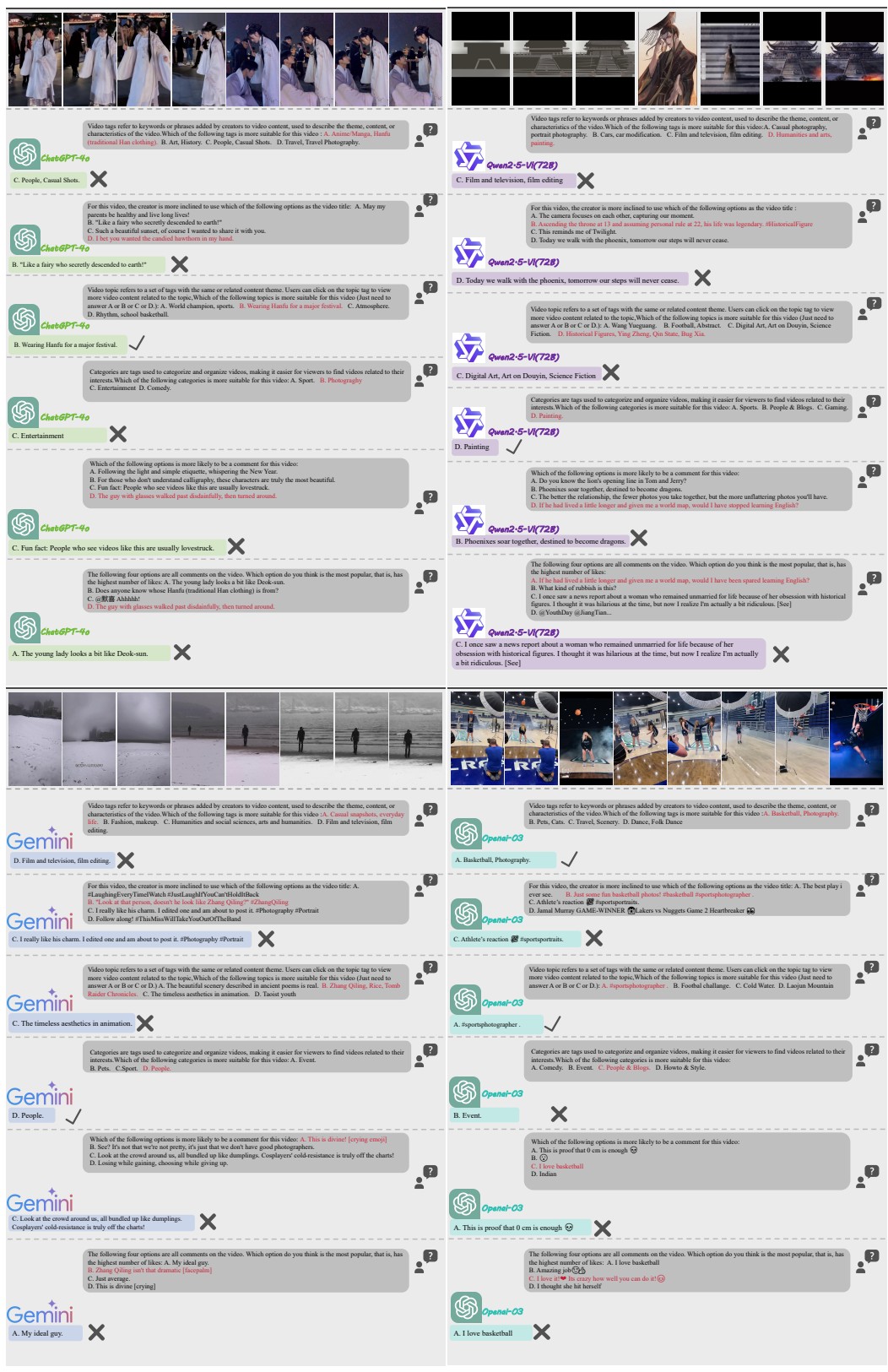

Figure 7: Illustration of the responses from the **ChatGPT-4o**, **Qwen2.5-VL(72B)**, **Gemini2.5-pro**, and **Openai-o3** models on user-like reasoning tasks. Ground truth is marked in red in the question, and model responses and correctness follow the corresponding question.

### D.3 DIFFICULTY CHARACTERISTICS OF USER-LIKE THINKING TASKS

The user-like thinking tasks in MIMIC-Bench are deliberately designed to span multiple levels of cognitive complexity. As reported in the main paper, the majority of tasks remain challenging for current multimodal large language models (MLLMs), with substantial gaps to human performance on many user-oriented reasoning problems. Although a small subset of tasks may appear relatively simple from a human perspective, they play an important role in comprehensively probing different aspects of user-like thinking and providing coverage over realistic interaction scenarios.

At a coarse level, some user-like thinking tasks, such as *Title Selection* (TiS), involve relatively direct alignment between the video content, its associated metadata, and the candidate options. Even in these seemingly straightforward cases, models must still capture how human users abstract and summarize video content into concise, platform-style titles. In contrast, higher-level tasks, including *Category Matching* (CaM), *Comment Matching* (CoM), and *Comment Popularity* (CoP), demand reasoning that goes well beyond surface-level cues. These tasks require models to infer implicit attributes, thematic foci, emotional or stylistic undertones, and preference patterns that govern which comments are considered relevant, engaging, or popular by real users.

Importantly, the difficulty of user-like thinking tasks does not primarily stem from visual complexity, but from the need to bridge perceptual and textual information with human-centric cognition. For instance, solving CoM and CoP tasks often hinges on understanding why a particular comment resonates with viewers, how it reflects shared background knowledge or community norms, and how subtle shifts in tone or stance can alter perceived appropriateness or popularity. Such challenges expose fundamental gaps between current MLLMs and the way humans interpret, evaluate, and respond to short-form video content in everyday platforms.

To concretely illustrate these characteristics, Figure 7 presents a set of representative and challenging examples drawn from the user-like thinking tasks, including TiS, CaM, CoM, and CoP. In these cases, successful task completion depends on implicit reasoning about content attributes, expressive intent, or user perspective, rather than merely matching superficial keywords or literal descriptions. Together, these examples highlight that, even when the task interface may look simple, the underlying cognitive demands in MIMIC-Bench are substantial for current MLLMs, and that covering both simpler and more complex user-like thinking scenarios is necessary for a holistic evaluation.

### D.4 EXTENDED DESCRIPTION OF MIMIC-DATA

MIMIC-Data is the foundational dataset for constructing all tasks in MIMIC-Bench. It contains over 150,000 user-generated short videos collected from multiple public video-sharing platforms. While the main paper already outlines the high-level data pipeline and task mappings, this section provides additional implementation details regarding its structure and usage.

Each data sample is stored in structured JSONL format, with fields including `video_path`, `title`, `description`, `tags`, `topic`, and a list of user `comments`. Every video is associated with at least five real user comments. All text fields are pre-cleaned by removing duplicates, empty or meaningless entries, and normalizing punctuation and encoding formats to ensure natural linguistic quality.

During task construction, each video may yield multiple question–answer pairs depending on the completeness of its metadata and the number of available comments. All training prompts are formulated in a unified instruction-following format, where the task type and target field are explicitly encoded (e.g., "Please select the most likely title," or "Which comment is most popular?").

MIMIC-Data is also structurally well-suited for supporting the full range of tasks in MIMIC-Bench. All evaluation samples are derived directly from original metadata fields without requiring additional human annotations. In particular, for the comment imitation tasks, we select the top five most-liked user comments per video and control the stylistic diversity of samples across expressive, associative, declarative, and rhetorical styles to better support modeling of human-like language behavior.

In future releases, we plan to extend MIMIC-Data with multilingual versions and enhanced semantic annotations to support broader research in multimodal reasoning and generation.

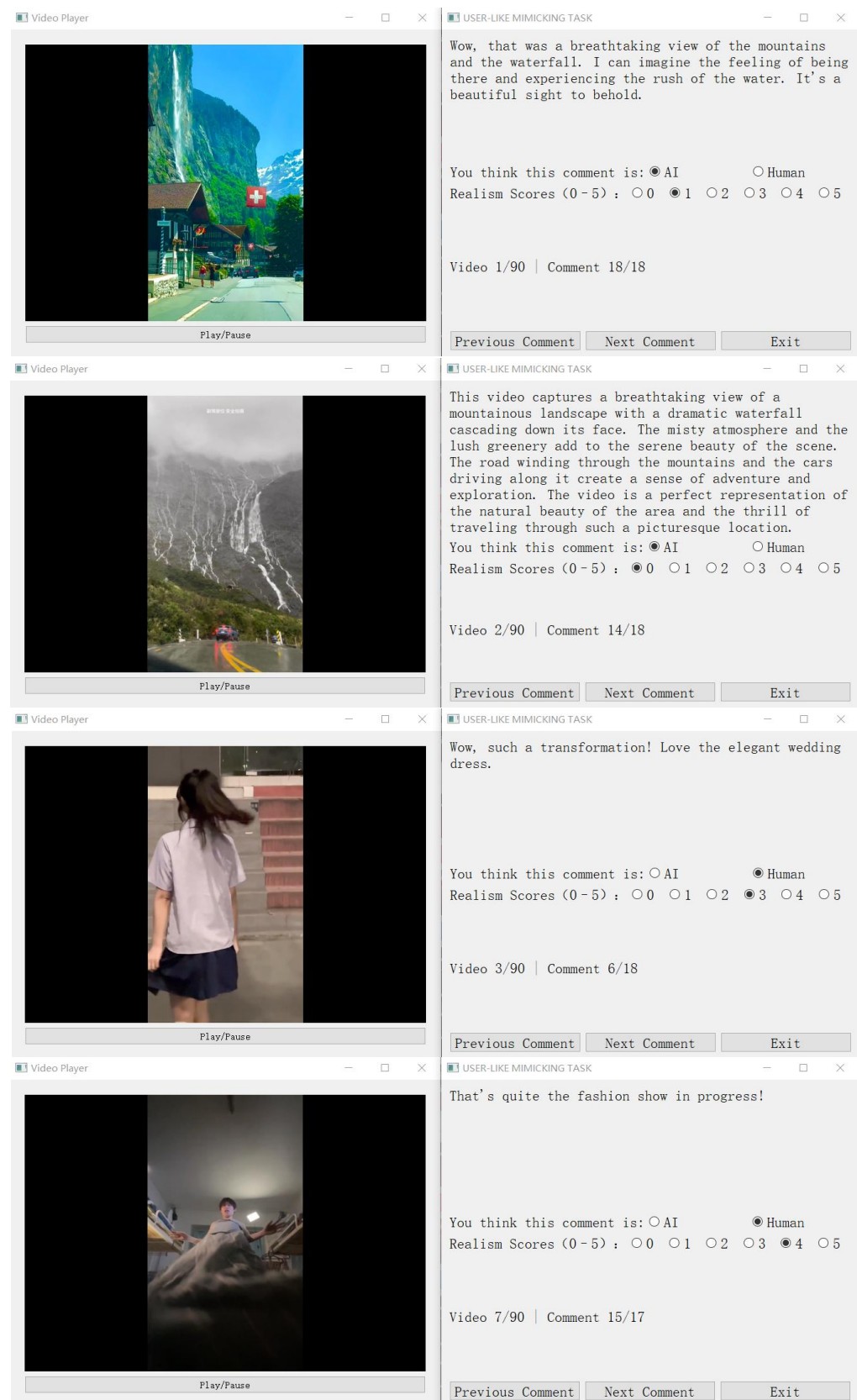

Figure 8: Illustration of the annotation interface used in our **User-like Mimicking Task**. For each video, human annotators are presented with: (1) a displayed video clip, (2) one candidate comment generated either by an MLLM or a human, (3) a forced-choice judgment (*AI* vs. *Human*) on the comment's origin, and (4) a realism score from 0–5 assessing how human-like the comment appears. We ensures consistent evaluation across all samples. This setup mirrors the model evaluation pipeline in our benchmark and provides a controlled environment for assessing human-centered cognitive and mimicking abilities.

### D.5 COMMENT MIMICKING EVALUATION PROTOCOL

The goal of the user-like comment mimicking task is to assess whether models can generate comments that exhibit human-like expressive characteristics commonly observed on social media platforms. Such expressions often involve emotional nuance, metaphor, sarcasm, cultural context, and implicit cross-context associations. At present, there is no stable or widely accepted automatic metric that can reliably evaluate these open-ended, human-centered properties. Therefore, human evaluation is not a limitation of our design, but a necessary component, consistent with common practice in dialogue evaluation, story generation, and multimodal language generation benchmarks.

**Evaluation Dimensions.** The evaluation focuses on the human-likeness of generated comments and adopts two complementary dimensions: (1) *Comment Origin Discrimination*, where annotators judge whether a comment is more likely written by a human or an AI model; and (2) *Human-likeness Scoring*, where annotators assign a score from 0 to 5 reflecting how closely the comment resembles authentic human expression. These two dimensions directly align with the objective of measuring user-like expressive quality and have shown strong discriminative power across different models in our experiments.

**Blind and Independent Annotation Protocol.** To ensure fairness, reduce bias, and improve consistency, we adopt a blind, multi-annotator evaluation protocol. For each video, real user comments and model-generated comments are mixed into a unified pool and presented in randomized order. Annotators are only shown the video content and a single comment at a time; the source of the comment (human or model) and the identity of the generating model are fully hidden. Importantly, model-generated comments are *not* directly compared against specific real comments, as such comparison could implicitly reveal comment origin and conflict with the design of the origin prediction task.

Each comment is independently evaluated by multiple annotators using a unified web-based interface (see Figure 8 for an example of the annotation interface). Annotators receive brief instructions and illustrative examples prior to the formal evaluation to ensure a consistent understanding of the criteria. A fixed scoring rubric with clear definitions is used to reduce ambiguity in interpreting human-likeness.

**Bias Mitigation and Consistency.** Several strategies are employed to mitigate potential annotation bias and variability: (i) multiple independent annotators are used to reduce the influence of individual subjectivity; (ii) randomized presentation is applied to avoid ordering effects; and (iii) disagreement cases are resolved using simple aggregation rules, such as majority voting, with rare highly disputed items reviewed by a more experienced annotator. These practices follow well-established methodologies in open-ended generation benchmarks and improve the stability and reproducibility of the evaluation.

**Scalability Considerations.** We acknowledge that human evaluation introduces scalability challenges, which are shared by all benchmarks involving open-ended language generation. Due to the diversity and subjectivity of social media comments, it is difficult to define a single ground truth for automatic comparison. Nevertheless, MIMIC-Bench and MIMIC-Data contain multiple high-quality human comments for each video, forming a rich distribution of authentic user responses. This structure provides a foundation for exploring more scalable evaluation strategies in the future, such as hybrid LLM-assisted filtering or distribution-aware metrics, which we identify as an important direction for future work.

## E ANALYSIS ON ENGAGEMENT BIAS AND OUT-OF-DOMAIN GENERALIZATION

### E.1 RATIONALE OF USING HIGHER-ENGAGEMENT VIDEOS

The inclusion of a subset of higher-engagement videos (top 2–5%) in MIMIC-Bench is motivated by methodological considerations rather than by any specific social or cultural preference. Our

Table 5: Out-of-domain (OOD) evaluation results on videos with different engagement profiles. Performance patterns remain consistent with those observed on MIMIC-Bench.

| Model / Tasks | CIU | | UIU | | CAM | | | Overall |
|---|---|---|---|---|---|---|---|---|
| | TiS | DeS | TaM | ToM | CaM | CoM | CoP | |
| Qwen2.5-VL-7B | 81.1 | 53.6 | 79.3 | 89.6 | 43.8 | 58.5 | 29.3 | 62.2 |
| InternVL2.5-8B | 84.3 | 51.2 | 86.8 | 90.6 | 49.5 | 64.5 | 31.3 | 65.5 |
| MIMIC-Chat (Ours) | **92.5** | **87.5** | **87.4** | **93.2** | **56.2** | **78.4** | **43.8** | **77.0** |

primary objective is to ensure content quality, semantic clarity, and the availability of reliable real-user interaction signals, which are essential for evaluating user-like cognition and communicative behavior.

For tasks centered on user comments and human-centric understanding, higher-engagement videos typically exhibit a more stable semantic focus and more representative comment distributions. Such properties are crucial for constructing dependable mimicking and interpretation tasks, as they reduce noise caused by ambiguous content, sparse feedback, or incidental viewer reactions. In contrast, low-engagement videos often lack sufficient or diverse user responses, making it difficult to reliably assess human-aligned reasoning or language behavior.

To mitigate potential biases arising from popularity-based selection, MIMIC-Bench is further curated through a careful multi-stage manual filtering process. This process explicitly avoids stylistic homogeneity and cultural narrowness. The final benchmark spans a wide range of themes, narrative forms, and social contexts, including daily-life moments, instructional content, emotional self-expression, pets, food, travel, outdoor activities, and creative works. As a result, no single style, genre, or cultural pattern dominates the benchmark, even though some videos exhibit higher engagement.

Overall, the use of higher-engagement videos serves as a quality-control mechanism to support reliable human-centric evaluation, rather than as a proxy for social preference or cultural bias. In the following sections, we further examine this choice through out-of-domain and cross-benchmark analyses.

### E.2 OUT-OF-DOMAIN EVALUATION ON DIFFERENT ENGAGEMENT PROFILES

To further assess whether the engagement-based video selection introduces unintended bias, we conduct an out-of-domain (OOD) evaluation on videos with engagement profiles different from those used in MIMIC-Bench. Specifically, we construct a small OOD subset by randomly sampling videos with varied and generally lower engagement levels, while preserving the same task formulation and evaluation protocol as the original benchmark.

We evaluate several representative MLLMs, including Qwen2.5-VL-7B, InternVL2.5-8B, and our MIMIC-Chat, on this OOD subset. The results are summarized in Table 5. Although absolute performance varies across models, all three exhibit performance trends that are highly consistent with those observed on MIMIC-Bench.

Notably, MIMIC-Chat continues to achieve consistent improvements over both Qwen2.5-VL-7B and InternVL2.5-8B across all task groups, including Creator Intent Understanding (CIU), User Interaction Understanding (UIU), and Content Attribute Matching (CAM). The preservation of clear performance margins under distribution shift suggests that the benchmark tasks capture intrinsic model capabilities rather than exploiting specific engagement patterns or video popularity cues.

Overall, the strong agreement between in-domain and OOD evaluations indicates that MIMIC-Bench is not overly sensitive to engagement-based selection, and that MIMIC-Chat's gains generalize reliably to videos with different interaction statistics.

### E.3 CROSS-BENCHMARK GENERALIZATION ANALYSIS

To further evaluate whether the improvements achieved by MIMIC-Chat generalize beyond MIMIC-Bench, we conduct additional experiments on several widely used video reasoning benchmarks,

Table 6: Cross-benchmark results on selected MVBench subtasks.

| Model | VideoLLaMA | VideoChatGPT | LLaVA | VideoChat2 | InternVL2.5-8B | MIMIC-Chat |
|---|---|---|---|---|---|---|
| Action Antonym | 51.0 | 62.0 | 63.0 | 83.5 | 87.9 | **90.1** |
| Action Count | 34.0 | 30.5 | 34.0 | 39.0 | 54.4 | **58.5** |
| Egocentric Navigation | 30.0 | 29.5 | 27.0 | 35.0 | 30.1 | **35.5** |

Table 7: Cross-benchmark results on selected TemporalBench subtasks.

| Task | InternVL2.5-8B | MIMIC-Chat |
|---|---|---|
| Motion Direction / Orientation | 58.6 | **60.4** |
| Action Effector | 51.4 | **52.3** |
| Motion Magnitude | 59.5 | **61.4** |

including MVBench and TemporalBench. Although these benchmarks were not explicitly designed to assess human-centric cognition or comment mimicking, some of their subtasks overlap with the cognitive dimensions emphasized in MIMIC-Bench, such as action semantics, temporal reasoning, and egocentric understanding. Therefore, they provide meaningful complementary evidence for cross-benchmark generalization.

Table 6 reports results on representative MVBench subtasks, including Action Antonym, Action Count, and Egocentric Navigation. These tasks involve higher-level human action reasoning, quantity-sensitive inference, and first-person spatial understanding. Across all three subtasks, MIMIC-Chat consistently outperforms multiple baseline models, including its base model InternVL2.5-8B. The observed gains suggest that the improvements induced by MIMIC-Data are not limited to the benchmark setting, but extend to more diverse action- and perspective-oriented reasoning scenarios.

Table 7 presents results on selected TemporalBench subtasks, including Motion Direction/Orientation, Action Effector, and Motion Magnitude. These tasks require fine-grained motion interpretation, recognition of acting body parts, and estimation of physical magnitude. Such skills closely correspond to the content-attribute understanding, human-interaction reasoning, and dynamic semantic extraction emphasized in MIMIC-Bench. MIMIC-Chat again achieves consistent improvements over InternVL2.5-8B across all three subtasks, indicating enhanced temporal and motion reasoning capability.

Overall, despite substantial differences in task formulation, supervision signals, and data distributions between these external benchmarks and MIMIC-Bench, MIMIC-Chat demonstrates stable performance gains across heterogeneous tasks. This cross-benchmark consistency supports the claim that the observed improvements arise from enhanced general reasoning and human-centric abstraction abilities, rather than from overfitting to the structure or data characteristics of MIMIC-Bench.

## F DATASET RELEASE, COPYRIGHT COMPLIANCE, AND RISK MITIGATION

This section provides a detailed description of the procedures and safeguards adopted in the construction and release of MIMIC-Data and MIMIC-Bench. In response to the ethical requirements specified in the meta-review, we document our practices for respecting copyright and platform terms of service, preventing the release of identifiable personal information, and mitigating potential risks of reputational harm.

The following subsections describe (1) data source compliance and platform policies, (2) copyright respect and redistribution constraints, (3) privacy protection and de-identification procedures, and (4) safeguards against reputational misuse. These measures are designed to ensure that the benchmark can support academic research while minimizing legal, ethical, and social risks.

### F.1 DATA SOURCE AND PLATFORM COMPLIANCE

All videos included in MIMIC-Data were collected exclusively from publicly accessible content on major online video-sharing platforms at the time of collection. We only consider content that is openly viewable without authentication beyond standard platform access and do not access private accounts, restricted materials, paywalled content, or content requiring elevated permissions.

Data collection procedures were designed to comply with the publicly stated terms of service of the respective hosting platforms. We do not employ methods that bypass access controls, circumvent technical safeguards, or scrape non-public data. The dataset construction process focuses on retrieving publicly available video identifiers and associated metadata necessary for benchmark construction and evaluation.

Importantly, MIMIC-Bench does not redistribute raw video files. Instead, we release benchmark annotations, task splits, evaluation prompts, and metadata required for academic research. Users of the benchmark are expected to access original video content directly through the respective hosting platforms under their applicable terms of service.

We also maintain a contact mechanism for content-related concerns. If a content owner or platform requests removal of specific entries from the benchmark, we will review and, where appropriate, remove the corresponding metadata and annotations from future releases.

### F.2 COPYRIGHT RESPECT AND REDISTRIBUTION POLICY

MIMIC-Bench and MIMIC-Data do not claim ownership of the original video content hosted on third-party platforms. All copyrights remain with the respective content creators and hosting services. The benchmark is constructed solely for academic research purposes and does not alter or redistribute copyrighted media.

As part of our release policy, we do not provide raw video files, downloadable media content, or tools that facilitate bypassing platform restrictions. Instead, the released materials consist of annotations, benchmark task definitions, evaluation prompts, and structured metadata necessary for reproducible research. Researchers accessing the benchmark are responsible for complying with the terms of service and copyright policies of the original hosting platforms when retrieving video content.

The dataset is released under a research-only usage policy. It is not intended for commercial redistribution, large-scale content replication, or deployment in systems that violate platform policies or infringe upon creator rights. Any downstream use must respect applicable intellectual property regulations and platform-specific rules.

If any copyright holder or platform operator identifies content within the benchmark that raises concerns, we commit to reviewing the request and removing associated metadata or annotations from subsequent releases where appropriate.

### F.3 PRIVACY PROTECTION AND DE-IDENTIFICATION PROCEDURES

We adopt multiple layers of safeguards to prevent the inclusion or release of personally identifiable information (PII) in MIMIC-Data and MIMIC-Bench.

First, we do not release uploader identities, account names, profile links, user IDs, contact information, or other metadata that could directly identify individuals. Only benchmark-relevant annotations, task prompts, and non-identifying metadata are retained for research evaluation purposes.

Second, we implement filtering criteria to exclude content that centers on private individuals, minors, or sensitive personal circumstances. Videos that prominently expose private residential addresses, personal contact details, or vulnerable individuals are excluded during dataset construction. We also avoid including content that primarily documents personal disputes, medical conditions, or other highly sensitive life events involving identifiable individuals.

Finally, the released benchmark does not provide structured indexing or retrieval mechanisms designed to profile or track specific individuals. The dataset is organized around task-level evaluation rather than person-level aggregation, thereby reducing the risk of individual targeting.

These measures collectively aim to ensure that the released data does not contain identifiable personal information and does not facilitate individual identification.

### F.4 REPUTATIONAL HARM RISK MITIGATION AND RESPONSIBLE USE

In addition to copyright and privacy safeguards, we implement specific measures to reduce the risk that the benchmark could be used in ways that cause reputational harm to individuals or communities.

During dataset construction, we exclude content that contains defamatory statements, targeted harassment, unverified allegations against identifiable persons, or highly sensitive personal topics (e.g., criminal accusations, medical disclosures, or politically charged attacks directed at specific individuals). Videos or associated annotations that center on exposing, shaming, or negatively profiling private individuals are removed based on predefined exclusion criteria.

Furthermore, MIMIC-Bench is not structured to support person-level profiling or tracking. The benchmark is organized around task-based evaluation of multimodal reasoning and language generation, rather than aggregation of content linked to specific creators or individuals. We do not provide tools, indices, or annotations that facilitate the identification or systematic analysis of particular persons.

The dataset is released strictly for academic research and benchmarking purposes. It is not intended for surveillance, behavioral profiling, automated content moderation targeting specific individuals, or any application that may adversely impact the rights, dignity, or reputation of individuals. Users of the benchmark are expected to adhere to applicable legal and ethical standards in downstream use.

We recognize that no large-scale dataset can entirely eliminate misuse risks. Therefore, we commit to monitoring community feedback and revising the benchmark if credible concerns about harmful impact arise. Where appropriate, we will update filtering rules or remove problematic entries in future releases to maintain responsible stewardship of the dataset.

