# OpenReview forum: "MIMIC-Bench: Exploring the User-Like Thinking and Mimicking Capabilities of Multimodal Large Language Models"
_ICLR.cc/2026/Conference — ICLR 2026 Poster_

### Official Review · Reviewer_wUW5 · 2025-10-27

**Soundness:** 4
**Presentation:** 4
**Contribution:** 4
**Rating:** 8
**Confidence:** 4

**Summary:**

To emulate user-like thinking and behaviour, the paper introduces MIMIC-Bench, a large-scale benchmark for evaluating human-like reasoning and social cognition in multimodal large language models (MLLMs). Built on MIMIC-Data—a dataset of over 150K user-generated videos with rich metadata such as titles, tags, and comments—MIMIC-Bench emphasizes how people think, feel, and react to videos, rather than only recognizing visual content.  It includes two main tasks: (1) User-like Thinking, covering creator intent, content attributes, and user interactions; and (2) User-like Mimicking, assessing models’ ability to generate human-like comments. The authors also develop MIMIC-Chat, a multimodal model trained on MIMIC-Data to jointly learn video semantics and human-style reasoning. Experiments on 24 state-of-the-art MLLMs show limited performance, while MIMIC-Chat demonstrates improved human-aligned understanding.

**Strengths:**

1. The explanation of the problem and experimental protocol is clear, and the figures and tables are well-organized, effectively helping readers understand both the problem and the model architecture.
2. The task is interesting and has clear real-world applications. Given a video, the model can generate human-like emotional responses and categorize the content into relevant tags. Such capabilities are essential for video understanding, predicting audience feedback on movies or online videos, and improving video recommendation systems.
3. The paper presents comprehensive experiments comparing the performance of multiple models on the proposed dataset.
4. For this task, the authors also propose MIMIC-Chat to estimate human responses. Based on the results, it serves as a strong and effective baseline.

**Weaknesses:**

1.  There are a few formatting issues—for example, in Lines 83–84, some words are incorrectly bolded, and in Lines 261–264, two metrics are presented but only one is highlighted in bold.
2. The evaluation procedure for the User-like Mimicking task is unclear. Is the generated response from each of the 24 MLLMs individually compared against the real user comments?
3. It would be helpful to include further insights and analysis based on the current models’ results—for example, whether the lower performance stems from limited perceptual ability or insufficient common-sense knowledge.

**Questions:**

None

---

> ### Author Response · Authors · 2025-11-26
> **Responses to Reviewer wUW5**
>
> **To reviewer wUW5:** Thanks for the constructive comments. Responses to specific comments are given below, which have also been incorporated into the revised paper.
>
> 1. **W1:** Formatting issues
>
>    Thanks for pointing out the formatting inconsistencies (e.g., incorrect bolding in Lines 83-84 and Lines 261-264). We have corrected all these formatting issues in the revised paper. Moreover, further proofreading is carefully performed to ensure overall consistency and prevent similar formatting errors throughout the manuscript.
>
> 2. **W2:** Clarity of the evaluation procedure for the User-like Mimicking task
>
>    Thanks for the questions. We should clarify that the generated comments from each of the 24 MLLMs are **not directly compared to real user comments**, since directly comparison may implicitly reveal the comment’s origin and conflict with the design of the comment origin prediction task. Instead, our evaluation follows a **blind and independent** protocol as follows. (1) Each model independently generates one comment per video. (2) Human annotators see the video and a mixed pool of comments (multiple human comments + multiple model-generated comments) one-by-one with random order, but **the source is hidden**. (3) Annotators judge whether a comment is human or AI (Comment Origin Prediction), and its human-likeness from a 0-5 scale. (4) All models follow the *same* generation and evaluation pipeline, ensuring fairness and comparability. We have included a concise description of this procedure in the revised version to improve clarity.
>
> 3. **W3:** Further insights and analysis
>
>    Thanks for the suggestions. We have included more insights and analysis in the revised manuscript as follows, particularly regarding whether the lower performance stems from limited perceptual ability or insufficient common-sense knowledge. As shown in Table 1, current MLLMs achieve relatively higher performance on TiS and ToM tasks, while showing lower performance on CaM and CoP tasks. Since TiS and ToM tasks primarily measure perceptual ability, while CaM and CoP tasks focus on evaluating higher-level common-sense or social-cognitive reasoning capabilities, this performance discrepancy suggests that the limitations of current models are less related to perceptual deficiencies and more closely tied to challenges in social-cognitive reasoning, such as interpreting emotional intent, implicit social cues, cultural context, sarcasm, and psychological projection. These capabilities require human-like inference beyond literal content, and remain challenging for existing models. Moreover, as shown in Table 2, current MLLMs exhibit a substantial performance gap compared to human performance, which further highlights their limitations in acquiring and applying human-like knowledge.
>
>     However, the performance ranking of the models remains relatively consistent across different tasks, indicating that more powerful models generally possess both stronger perceptual abilities and common-sense knowledge.

---

### Official Review · Reviewer_cT9q · 2025-10-30

**Soundness:** 2
**Presentation:** 2
**Contribution:** 2
**Rating:** 4
**Confidence:** 4

**Summary:**

This paper presents **MIMIC-Bench**, a large-scale benchmark for human-aligned video understanding, together with **MIMIC-Data**, a supporting dataset, and **MIMIC-Chat**, a model fine-tuned for human-like reasoning and comment generation. The benchmark consists of two complementary tracks—User-like Thinking Tasks (structured reasoning based on metadata) and User-like Mimicking Tasks (comment imitation). The authors evaluate 24 multimodal large language models (MLLMs), including both open-source and proprietary systems, and provide human evaluations for reference.

**Strengths:**

1. The creation of MIMIC-Data and MIMIC-Bench fills a gap in evaluating human-aligned reasoning in video understanding, moving beyond visual recognition to social and cognitive aspects.
2. The benchmark incorporates structured reasoning tasks and a creative comment imitation task, encouraging evaluation of both perceptual and cognitive understanding.
3. The dual-branch spatiotemporal encoding and unified instruction format in MIMIC-Chat demonstrate careful design and solid engineering implementation.
4. Extensive experiments on a large set of baselines and human participants give a comprehensive picture of current model capabilities.

**Weaknesses:**

1. Since MIMIC-Chat is trained on MIMIC-Data, which shares sources with the benchmark,  there is potential concern regarding content overlap or stylistic memorization.
2. MIMIC-Bench focuses on highly engaging videos (top 2–5%), which might bias the content toward certain social or cultural styles. The work would benefit from analysis or experiments on more diverse or out-of-domain video data.
3. While the Comment Imitation task benefits from human judgment, this can lead to variability in evaluation. Including more information on the annotation process, strategies for mitigating bias, and possible alternative metrics would help improve reproducibility and clarity.

**Questions:**

1. Could the authors elaborate on whether there is any potential overlap between MIMIC-Data (used for training) and MIMIC-Bench (used for evaluation)? If so, how is such overlap avoided or controlled to ensure fair assessment?

2. It would be helpful if the authors could describe the human annotation process for comment imitation in greater detail—e.g., annotation interface, annotator expertise, and disagreement resolution.

---

> ### Author Response · Authors · 2025-11-26
> **Responses to Reviewer cT9q (Part I)**
>
> **To reviewer cT9q:** Thanks for the constructive comments. Responses to specific comments are given below, which have also been incorporated into the revised paper.
>
> 1. **W1 & Q1:** Potential concern regarding content overlap or stylistic memorization between MIMIC-Data and MIMIC-Bench
>
>     Thanks for the concern. It should be clarified first that MIMIC-Bench and MIMIC-Data are strictly disjoint. The 4,000 benchmark videos were selected first through an independent filtering process and were explicitly excluded from the training split, i.e., MIMIC-Data. No benchmark videos and its associated metadata appears in MIMIC-Chat’s training set, eliminating any content overlap and leakage.
>
>     Regarding stylistic memorization, several factors mitigate this risk. (1) Although both datasets originate from the same broad distribution of user-generated videos, the content in MIMIC-Data is highly diverse, spanning a wide range of topics, cultural backgrounds, video formats, and interaction styles. Such diversity makes it unlikely for a model to memorize a narrow stylistic pattern that would transfer directly to benchmark tasks. (2) In addition, similar to most of the training data for MLLMs, MIMIC-Data contains large-scale captured online user-generated social videos with raw, unprocessed media data (such as title, tag, comments, etc.), while MIMIC-Bench is a curated evaluation benchmark with manually designed and refined QA pairs. This mitigates the stylistic memorization. The improvement of MIMIC-Chat precisely reflect that unprocessed online user-generated social videos and their corresponding data can improve the user-like thinking and mimicking capabilities, and current MLLMs have not seen this kind of data and lack corresponding knowledge.
>
>     We appreciate the reviewer’s attention to this issue and have made the data construction and separation protocol clearer in the revised version.

---

> ### Author Response · Authors · 2025-11-26
> **Responses to Reviewer cT9q (PartⅡ)**
>
> 2. **W2:** Higher-engagement videos (top 2–5%) may introduce social or cultural bias, analysis on out-of-domain data.
>
>     Thanks for the comments and suggestions. First of all, we would like to clarify the rationale of using higher-engagement videos in MIMIC-Bench and how we mitigate potential bias. (1) Although the benchmark contains a portion of highly engaging videos, this selection is not driven by any particular social or cultural preference. Instead, it is motivated by the need to ensure content quality, semantic clarity, and reliable real-user interaction signals. For tasks involving user comments and human-centric cognition, higher-engagement videos typically exhibit more stable semantic focus and more representative comment distributions, which are important for constructing dependable mimicking and interpretation tasks. (2) MIMIC-Bench was constructed through careful multi-stage manual filtering to avoid stylistic homogeneity or cultural narrowness. The final benchmark covers a wide range of themes, narrative forms, and social contexts, such as daily life, how-to content, emotional expression, pets, food, travel, outdoor activities, creative works, and more, ensuring no single style or culture dominates, even when some videos have higher engagement.
>
>    We agree that further enhancing diversity in terms of long-tail topics, cross-cultural content, and varied engagement levels is important. To this end, we have additionally constructed a small out-of-distribution (OOD) subset consisting of randomly sampled videos with different engagement profiles. Due to time constraints, we have currently evaluated several representative MLLMs and our MIMIC-Chat on this subset, and we will extend the evaluation to the full model suite and release the complete results in the camera-ready version. The results are given below, and it can be observed that all three models exhibit consistent performance patterns on this out-of-distribution (OOD) subset compared to the results reported on MIMIC-Bench. Specifically, our MIMIC-Chat model continues to outperform both Qwen2.5-VL-7B and InternVL2.5-8B across all task groups, yielding a clear and stable performance margin even when the data distribution differs from the benchmark. This strong agreement between in-domain and OOD evaluations indicates that the task design of MIMIC-Bench reliably captures intrinsic model capabilities rather than capturing particular video styles or engagement levels.
>
>     |Task Type|CIU||UIU|||CAM||overall|
>     |---------|---|---|---|---|---|---|---|-------|
>     |Model / Tasks|TiS|DeS|TaM|ToM|CaM|CoM|CoP| |
>     |Qwen2.5-VL-7B|81.1|53.6|79.3|89.6|43.8|58.5|29.3|62.2|
>     |InternVL2.5-8B|84.3|51.2|86.8|90.6|49.5|64.5|31.3|65.5|
>     |MIMIC-Chat(Ours)|92.5|87.5|87.4|93.2|56.2|78.4|43.8|77.0|
>
>     To further verify whether MIMIC-Chat exhibits cross-benchmark generalization, we have conducted additional evaluations on several widely used video reasoning benchmarks. Although these benchmarks were not designed for human centric cognition or comment mimicking capabilities, some of their subtasks overlap with the cognitive dimensions emphasized in MIMIC-Bench, such as action semantics, temporal reasoning, and egocentric understanding, and therefore serve as meaningful generalization supplements. The results are summarized below.
>
>     |MVBench\Model|VideoLLaMA|VideoChatGPT|LLaVA|VideoChat2|InternVL2.5-8B|MIMIC-Chat|
>     |-|--|--|--|--|--|--|
>     |Action Antonym|51|62|63|83.5|87.9|**90.1**|
>     |Action Count|34|30.5|34|39|54.4|**58.5**|
>     |Egocentric Navigation|30|29.5|27|35|30.1|**35.5**|
>
>     |TemporalBench\Model|InternVL2.5-8B|MIMIC-Chat|
>     |-|--|--|
>     |Motion Direction/Orientation|58.6|**60.4**|
>     |Action Effector|51.4|**52.3**|
>     |Motion Magnitude|59.5|**61.4**|
>
>     The MVBench subtasks, Action Antonym, Action Count, and Egocentric Navigation, involve higher-level human action reasoning, quantity sensitive inference, and first person spatial understanding, and TemporalBench subtasks, Motion Direction/Orientation, Action Effector, and Motion Magnitude, correspond to the content-attribute understanding, human-interaction reasoning, and dynamic semantic extraction emphasized in MIMIC-Bench. The performance gain over our base model InternVL2.5-8B, indicates our generalization on more diverse or out-of-domain video data.

---

> > ### Author Response · Authors · 2025-11-26
> > **Responses to Reviewer cT9q (PartⅢ)**
> >
> > 3. **W3 & Q2:** More information on the annotation process, strategies for mitigating bias
> >
> >    Thanks for the concern. The goal of the comment mimicking task is to assess whether models can generate comments with *human-like expressive characteristics*. Such comments often involve emotional nuance, metaphor, sarcasm, cultural context, and implicit associations. At present, there is no reliable automatic metric that can robustly capture these aspects in NLP/MLLM evaluation. Human judgment is therefore not a flaw but an essential component of this task, and is widely adopted in open-ended generation benchmarks (e.g., dialogue evaluation, creative writing, and social-media-style generation).
> >
> >    Before the formal experiment, we have designed detailed evaluation protocol to **reduce potential variability and bias**. Information on the annotation process and strategies are illustrated as follows. (1) **Multiple annotators:** Each comment is independently rated by several annotators to reduce individual bias. (2) **Blind evaluation:** The interface hides the comment source (model vs. human) and model identity. Annotators only see the video and the comment. (3) **Unified interface & brief training:** All annotators use the same interface and receive short instructions with examples to ensure consistent understanding of the criteria. (4) **Fixed scoring rubric:** Clear definitions and illustrative examples are provided to reduce ambiguity in interpreting “human-like”. (5) **Randomized presentation:** Real and generated comments are randomly mixed and shuffled to mitigate ordering effects. (6) **Disagreement handling:** Divergent cases are aggregated via simple consensus rules (e.g., majority vote), with rare disputed items reviewed by a more experienced annotator. These practices follow established methodology in open-ended generation benchmarks and improve stability and reproducibility. We have included a concise description of the annotation process, bias-mitigation steps, as well as experiment interface in the revised version.
> >
> >    Regarding possible alternative metrics would help improve reproducibility and clarity, due to the diversity and subjectivity of social media comments, it is hard to use one person's annotation as ground truth for evaluation. Despite the challenges, our MIMIC-Bench and MIMIC-Data contains multiple high-quality human comments per video, forming a rich distribution of authentic user responses. This creates the foundation for developing possible more scalable evaluation strategies (e.g., hybrid LLM-assisted filtering or distribution-aware metrics) in the future.

---

### Official Review · Reviewer_f8nN · 2025-11-01

**Soundness:** 2
**Presentation:** 3
**Contribution:** 3
**Rating:** 6
**Confidence:** 4

**Summary:**

This paper addresses the gap where existing MLLM benchmarks primarily focus on basic visual reasoning and neglect the ability of models to emulate user-like thinking when interpreting videos and their associated social context on platforms like social media. The authors propose a new dataset, a benchmark, and a specialized model. Experiments reveal MLLMs' limited capabilities in human-aligned thinking and responses.

**Strengths:**

- This paper proposes a fundamentally new evaluation paradigm, shifting the focus from basic visual reasoning to user-like thinking and mimicking capabilities for User-Generated Content
- This paper is well-written and easy to understand.

**Weaknesses:**

- The evaluation paradigm for the Mimicking Task (Comment Generation) suffers from poor scalability. Any new model introduced to the benchmark would necessitate re-evaluation, making comparative assessment difficult. Furthermore, the dimensions used for evaluation are insufficient.

**Questions:**

Please refer to weaknesses.

**Details Of Ethics Concerns:**

The reliance on User-Generated Content (UGC) harvested from social media platforms inherently raises significant ethical concerns regarding user privacy and data security.

---

> ### Author Response · Authors · 2025-11-26
> **Responses to Reviewer f8nN**
>
> **To reviewer f8nN:** Thanks for the constructive comments. Responses to specific comments are given below, which have also been incorporated into the revised paper.
>
> 1. **W1:** On the scalability and evaluation dimensions of the comment mimicking task
>
>     Thanks for the comments and concerns. We would like to clarify that the goal of the mimicking task is to evaluate whether models can reproduce the linguistic style, psychological expression, and social context characteristic of real users on social media. Such expressions often involve metaphor, emotional tension, sarcasm, cultural subtext, and cross-context associations, factors for which no stable, reliable automated metric currently exists. Therefore, human evaluation is not a limitation of our design but an inherent requirement for assessing this type of open-ended, human-centered generation, consistent with common practice in dialogue, story generation, and MLLM evaluation benchmarks.
>
>     Regarding scalability, we appreciate the reviewer’s thoughtful concern, but it is a general challenge shared by all benchmarks involving open-ended language generation. Some open benchmarks use LLM-based comparisons between human annotations and model outputs as their evaluation protocol, which assumes a relatively well-defined ground truth. However, social media comments are diverse and subjective, thus it is hard to use one person's annotation as ground truth for evaluation as aforementioned. Despite the challenges, our MIMIC-Bench and MIMIC-Data contains multiple high-quality human comments per video, forming a rich distribution of authentic user responses. This creates the foundation for developing more scalable evaluation strategies (e.g., hybrid LLM-assisted filtering or distribution-aware metrics), which we explicitly identify as an important direction for future extensions.
>
>     Regarding evaluation dimensions, It should be noted that the primary objective of the mimicking task is to measure human-likeness of generated content. The two core dimensions we adopt, “origin discrimination” and “human-likeness scoring”, directly align with this goal and have demonstrated strong discriminative power across different models. We agree that additional fine-grained dimensions can further enrich the evaluation, however, similar to the problem above, there is no robust and widely accepted method to reliably quantify these dimensions (e.g., stylistic diversity, emotional consistency, or pragmatic appropriateness) in a scalable and objective manner. We have therefore discussed these challenges and potential extensions in the revised manuscript and included them as part of our future work.

---

### Official Review · Reviewer_Y9c1 · 2025-11-01

**Soundness:** 3
**Presentation:** 2
**Contribution:** 2
**Rating:** 2
**Confidence:** 4

**Summary:**

This paper introduces MIMIC-Data, a large-scale dataset of over 150K user-shared videos with rich contextual metadata such as captions, tags, and comments. Building upon it, the authors present MIMIC-Bench, a benchmark of 4K videos designed to evaluate MLLMs’ ability to mimic human-like thinking and responses in realistic social media contexts. They further develop MIMIC-Chat, a model that integrates spatiotemporal understanding with fine-tuning for user-like reasoning and comment imitation. Experiments on 24 existing MLLMs show limited human-like behavior, while MIMIC-Chat demonstrates improved alignment with user-style cognition and response.

**Strengths:**

1. The paper nicely fills a research gap by being the first to focus on human-like thinking and imitation of MLLMs in user-generated video contexts. The proposed MIMIC-Data is large-scale and well-curated, offering rich multimodal metadata that will benefit future research. The MIMIC-Bench benchmark is also valuable for assessing real-world user-aligned reasoning.

2. The proposed MIMIC-Chat model is well-motivated — it integrates spatiotemporal features into an LLM and uses instruction tuning to balance structured reasoning with open-ended generation.

3. The experiments are thorough and convincing, comparing 24 state-of-the-art MLLMs (both open and closed-source) and incorporating human annotations as an upper bound, which strengthens the reliability of the results.

**Weaknesses:**

1. My main concern is that the core contribution — MIMIC-Bench — feels too simple. In Tables 1 and 2, existing models already achieve very high scores, and the benchmark cases shown (e.g., in Fig. 1) lack strong distractors, making the evaluation less convincing.

2. The paper does not report the video length distribution in MIMIC-Data. If short clips dominate, it may fail to assess long-video understanding. Also, the computational cost (e.g., GPU memory, inference speed) of MIMIC-Chat is not discussed, which limits understanding of its practical deployability.

3. The paper does not address potential overfitting — whether MIMIC-Chat overfits MIMIC-Data or whether MIMIC-Data and MIMIC-Bench are too similar. Moreover, results on other benchmarks are missing, leaving generalization unverified.

**Questions:**

Please refer to the Weaknesses section — the most critical point is to increase the difficulty of MIMIC-Bench to better match the rapidly improving capabilities of current VLMs.

---

> ### Author Response · Authors · 2025-11-26
> **Responses to Reviewer Y9c1 (Part I)**
>
> **To reviewer Y9c1:** Thanks for the constructive comments. Responses to specific comments are given below, which have also been incorporated into the revised paper.
>
> 1. **W1:** MIMIC-Bench appears too simple.
>
>     Thanks for the concern. First of all, we would like to clarify that most tasks in MIMIC-Bench are difficult for current MLLMs. While a small portion of tasks in MIMIC-Bench appear simple, it is important and necessary for comprehensively exploring the user-like thinking and reasoning capabilities of MLLMs. **(1) For user-like thinking and reasoning tasks**, as shown in Table 1, the overall performance for the best open-source model is 67.5%, for the best proprietary model is 75.1%, indicating that MIMIC-Bench is not “too simple”. Moreover, the performances of most tasks for the best open-source model, InternVL3, are around or below 75%, such as DeS, CaM, CoM, CoP, which indicates that current MLLMs have difficulties in human-like cognitive reasoning. Especially for CaM and CoP tasks, even the best proprietary models, including o3 and Gemini2.5-pro, perform poorly on these tasks. For TiS and ToM tasks, considering the ability of current MLLMs, it is reasonable that these models (especially proprietary models) perform well on these two tasks. However, as a comprehensive benchmark designed to capture the full spectrum of user-level human understanding, and benchmark user-like thinking capabilities, we still incorporate these two tasks to preserve the integrity of MIMIC-Bench. **(2) For the user-like mimicking task**, as shown in Table 2, almost all models perform poorly in terms of all metrics, indicating the difficulty of the task. As one of the core challenges of MIMIC-Bench, the mimicking task evaluates whether a model can achieve real user-generated expressive behavior. The results indicate that even the strongest closed-source models show a large gap compared to humans, which highlight the key bottlenecks of current MLLMs in human-like expression and underscore the long-term value of this task within the benchmark. **(3) Fig. 1 is an overview of the benchmark cases.** We show simple tasks to better illustrate the practical applications of our MIMIC-Bench. We have included more challenging tasks in Appendix Figure 7 to illustrate that most tasks are challenging for MLLMs in our benchmark. Moreover, the performance in Table 1 & 2 can also show the difficulties.
>
>     Moreover, MIMIC-Bench exhibits strong discriminative power across all tasks. Although a few closed-source models perform well on some common tasks, the overall performance gaps across the benchmark remain substantial: closed-source models consistently outperform mid-sized open-source models, which in turn outperform smaller models. Even different size variants within the same model family show clear performance stratification. This stable performance hierarchy demonstrates that MIMIC-Bench can effectively distinguish model capability, rather than yielding uniformly high scores.
>
> 2. **W2:** Regarding video-length distribution and computational cost
>
>     Thanks for the comments and suggestions. We have included the complete video-length statistics in the revised manuscript. The distribution is as follows: 1–10 s: 32.76%, 10–45 s: 49.08%, 45–180 s: 11.05%, and >180 s: 7.11%. These statistics indicate that while MIMIC-Bench mainly targets short-video scenarios which dominates real-world user-generated platforms, it still contains about 18% medium-to-long videos, with over 7% of these videos exceeding 3 minutes in length and the longest reaching 1 hour. This shows that the benchmark not only covers typical short-form content but also includes a meaningful proportion of longer clips, enabling a more comprehensive evaluation spectrum. To further contextualize this distribution, we refer to other widely used video benchmarks. MVBench is dominated by short clips, with most videos under 15 seconds. Meanwhile, Video-MME includes short, medium, and long videos, with durations ranging roughly from around 10 seconds to about one hour. Compared with these benchmarks, our distribution is more closely aligned with the real-world short-video ecosystem while still covering a reasonable amount of medium-length and longer videos, enabling evaluation across both short- and medium/long-term reasoning scenarios.
>
>     Regarding computational cost, MIMIC-Chat requires approximately 24 GB of GPU memory and processes a typical video (~20 seconds) in about 5–10 seconds. This resource footprint is comparable to mainstream multimodal models of similar scale and supports both research experiments and practical prototype deployment. We have added a concise summary of these computational requirements in the revised manuscript.

---

> ### Author Response · Authors · 2025-11-26
> **Responses to Reviewer Y9c1 (Part II)**
>
> 3. **W3:** Regarding potential overfitting and generalization
>
>     Thanks for the comments. It should be clarified that MIMIC-Bench and MIMIC-Data are strictly disjoint. The 4,000 benchmark videos were selected before constructing the training split, and none of these videos appear in the training set of MIMIC-Chat, eliminating any data leakage or memorization of benchmark content. Moreover, it should be noted that the motivation of MIMIC-Data is to enable the user-like thinking and mimicking capabilities of MLLMs. Thus, similar to most of the training data for MLLMs, MIMIC-Data contains large-scale captured online user-generated social videos with raw, unprocessed media data (such as title, tag, comments, etc.), while MIMIC-Bench is a curated evaluation benchmark with manually designed and refined QA pairs. The improvement of MIMIC-Chat precisely reflect that unprocessed online user-generated social videos and their corresponding data can improve the user-like thinking and mimicking capabilities, and current MLLMs have not seen this kind of data and lack corresponding knowledge.
>
>     Moreover, if overfitting were a dominating factor, MIMIC-Chat may achieve unusually high or near-saturated performance on MIMIC-Bench. However, as shown in Table 1, despite its significant improvements, MIMIC-Chat still does not surpass Gemini-2.5 Pro, a strong closed-source model, and substantial gaps remain on the most challenging cognitive reasoning tasks. This observation suggests that the gains do not stem from overfitting, but rather from enhanced generalization in user-like reasoning and understanding.
>
>
>     To further verify whether MIMIC-Chat exhibits cross-benchmark generalization, we have conducted additional evaluations on several widely used video reasoning benchmarks. Although these benchmarks were not designed for human centric cognition or comment mimicking capabilities, some of their subtasks overlap with the cognitive dimensions emphasized in MIMIC-Bench, such as action semantics, temporal reasoning, and egocentric understanding, and therefore serve as meaningful generalization supplements. The results are summarized below.
>
>
>     |MVBench\Model|VideoLLaMA|VideoChatGPT|LLaVA|VideoChat2|InternVL2.5-8B|MIMIC-Chat|
>     |-|--|--|--|--|--|--|
>     |Action Antonym|51|62|63|83.5|87.9|**90.1**|
>     |Action Count|34|30.5|34|39|54.4|**58.5**|
>     |Egocentric Navigation|30|29.5|27|35|30.1|**35.5**|
>
>
>     |TemporalBench\Model|InternVL2.5-8B|MIMIC-Chat|
>     |-|--|--|
>     |Motion Direction/Orientation|58.6|**60.4**|
>     |Action Effector|51.4|**52.3**|
>     |Motion Magnitude|59.5|**61.4**|
>
>     The MVBench subtasks, Action Antonym, Action Count, and Egocentric Navigation involve higher-level human action reasoning, quantity sensitive inference, and first person spatial understanding. As shown in the results, MIMIC-Chat consistently surpasses many open-source models, including our base model InternVL2.5-8B, on all three tasks, indicating that the improvements induced by MIMIC-Data extend beyond our own benchmark and support more robust human-action and perspective reasoning.
>
>     On TemporalBench, the subtasks Motion Direction/Orientation, Action Effector, and Motion Magnitude require fine-grained motion interpretation, recognition of the acting body part, and estimation of physical magnitude. These skills closely correspond to the content-attribute understanding, human-interaction reasoning, and dynamic semantic extraction emphasized in MIMIC-Bench. MIMIC-Chat again outperforms InternVL2.5-8B across all subtasks, showing stable improvements in temporal reasoning and motion understanding.
>
>     Overall, despite substantial differences in task formulation and data distribution between these external benchmarks and MIMIC-Bench, MIMIC-Chat demonstrates consistent and clear gains across multiple heterogeneous subtasks. This cross-benchmark consistency suggests that the improvements are not confined to the structure or style of MIMIC-Bench, but instead reflect enhanced capabilities in action semantics, temporal reasoning, and human-centric abstraction. These external results further support our conclusion that MIMIC-Chat’s performance gains arise from genuine generalization rather than overfitting to MIMIC-Bench.

---

### Author Response · Authors · 2025-11-26
**Summary of Our Responses & Paper Revisions**

We sincerely thank all reviewers for their valuable and constructive feedback. We appreciate that the reviewers recognized the significance of MIMIC-Bench, including its **clear motivation and timely relevance** (Reviewers Y9c1, cT9q), its **comprehensive multi-level task design** (Reviewers cT9q, wUW5), and the **soundness and strong presentation quality** of our work (Reviewer wUW5). We are also grateful that the reviewers acknowledged the novelty and importance of evaluating **human-like cognitive and social reasoning** in multimodal video understanding (Reviewers Y9c1, f8nN, cT9q).

We have made our best effort to address all concerns and improve the clarity of the paper. The major clarifications and revisions made in the updated manuscript are summarized as follows.

1. **Clarifying the difficulty and discriminative power of MIMIC-Bench:**
For the concerns about the simplicity of certain reasoning tasks. We have clarified that these tasks form only a small portion of the benchmark, which is used for enriching the benchmark to cover comprehensive user-like cases. Beyond these tasks, the majority of MIMIC-Bench require high-level human-like cognitive reasoning such as emotional inference, implicit intention, and comment-level abstraction, which are more difficult. We have highlighted the significant gap between humans and MLLMs on these challenging tasks (e.g., CoM/CoP), and the performance differences across 24 MLLMs. We have refined the description of task motivation and emphasized the cognitive difficulty of high-level tasks in the revised version. Moreover, besides reasoning evaluation, another contribution of MIMIC-Bench is to explore the mimicking capabilities as discussed in Table 2, which are more challenging than the reasoning tasks.

2. **Reporting video length distribution and clarifying MIMIC-Chat inference cost:**
Regarding concerns about video duration distribution and inference cost, we have provided the video length distribution and have clarified that MIMIC-Bench contains both short videos and long videos. We have also reported that MIMIC-Chat requires ~24GB GPU memory and takes ~5–10 seconds per 20s clip. These statistics will be added to the revised manuscript.

3. **Demonstrating the independence of MIMIC-Data and MIMIC-Bench and addressing overfitting concerns:**
For concerns about possible overlap or memorization, we have confirmed that MIMIC-Bench was constructed first through multi-stage filtering, and the remaining videos became MIMIC-Data; no video in the benchmark was used in model training. We have added clarifications regarding dataset separation and the distinct objectives of the two resources to Section 3.

4. **Clarifying the annotation process, bias mitigation, and consistency in the comment mimicking evaluation:**
Regarding concerns about more details on the annotation protocol and potential variability, we have explained our blind, multi-annotator setup, i.e., unified interface, standardized rubric, randomized presentation, and simple majority-vote disagreement resolution. We have added a concise description of this protocol in the revised paper.

5. **Clarifying the evaluation procedure of the User-like Mimicking task:**
Regarding clarification on how generated comments are evaluated, we have clarified that model-generated comments are not directly compared with real comments, since this would reveal their origin. Instead, annotators see a blind mixed pool (real + model comments) and independently judge comment origin and human-likeness. We have incorporated this clarification in the revised version.

6. **Providing additional insights into model failure modes:**
For further insights into why current models perform poorly on high-level semantic tasks, we have summarized that these failures come from limited social-cognitive reasoning and inability to infer human intent, emotion, and sociocultural cues beyond literal content. We also note that the multi-level design of MIMIC-Bench naturally offers diagnostic views across perceptual vs. higher-order reasoning failures. We have added a brief discussion to the revision.

7. **Addressing minor formatting issues:**
We have corrected the few misplaced boldface words in Lines 83–84 and 261–264, and have further proofread the paper.

---

### Author Response · Authors · 2025-12-02
**To Reassigned Area Chair**

We sincerely thank the new area chair taking the time to handleing our paper. To facilitate a smooth transition, we provide this concise summary to assist you in quickly understanding the paper and its current review status.

In this work, we present MIMIC-Bench, the first benchmark that systematically evaluates multimodal large language models on human centered video understanding, spanning 4,000 real user-generated videos and seven reasoning tasks covering creator intent, content attribution, and user interaction understanding. Beyond reasoning, we further introduce a new human-like mimicking task that assesses whether models can generate and discriminate authentic user-style comments. Evaluations over 24 leading MLLMs reveal substantial gaps from human performance and clear stratification across model families. Finally, we develop MIMIC-Chat, a model fine-tuned on MIMIC-Data that significantly improves both user-like cognitive reasoning and human-like comment expression, supported by detailed analyses, cross-benchmark generalization, and newly added OOD validation.

Below, we summarize the key concerns raised across the reviewers and how we have addressed them through targeted clarifications, additional analyses, and corresponding revisions during the rebuttal phase.

1. **Task Difficulty & Benchmark Discriminative Power:**
   We clarified that only a small subset of tasks in MIMIC-Bench is simple, while the majority require high-level cognitive and social reasoning such as emotional inference, implicit intention, and comment-level abstraction. We highlighted the substantial human–model gaps and stable performance stratification across 24 MLLMs, demonstrating strong discriminative power.

2. **Video-Length Distribution & Computational Cost:**
   We provided complete video-length statistics showing that the benchmark contains both short videos and a meaningful proportion of medium-to-long videos. We also reported the inference memory and latency of MIMIC-Chat (~24GB GPU memory and 5–10 seconds per 20s clip), addressing the reviewers’ concerns about computational requirements.

3. **Data Separation, Overfitting Concerns & Role of MIMIC-Data:**
   We clarified that MIMIC-Bench and MIMIC-Data are strictly disjoint, with the benchmark constructed first and fully excluded from training. We further explained the distinct purposes of the two resources and showed that MIMIC-Chat still exhibits substantial gaps to strong proprietary models, indicating that the improvements reflect enhanced generalization rather than memorization.

4. **Annotation Reliability & Mimicking Evaluation Protocol:**
   We provided a detailed description of the blind multi-annotator protocol—unified interface, standardized rubric, randomized presentation, and majority-vote resolution—to address concerns about annotation rigor. We also clarified the mixed-pool blind evaluation procedure for the User-like Mimicking task and explained why human assessment is necessary for open-ended human-likeness evaluation.

5. **Failure Modes & Cross-Benchmark/OOD Generalization:**
   We added analysis showing that model limitations mainly arise from social-cognitive reasoning rather than perceptual ability. We further conducted additional evaluations on MVBench, TemporalBench, and an out-of-distribution subset, demonstrating that MIMIC-Chat’s improvements generalize beyond MIMIC-Bench.

Beyond these major concerns, we have provided point-for-point responses to all remaining reviewer comments and completed all revisions by the rebuttal deadline. Before the score rollback, no reviewers engaged further in discussion or updated their scores. All major revised contents are highlighted in blue in the updated manuscript.

---

### Meta-Review · Program_Chairs · 2026-01-09

**Summary:**

This paper presents MIMIC-Bench, a human-centered video understanding benchmark, MIMIC-Data, a large-scale dataset containing 150K+ user-shared videos, and MIMIC-Chat, a model fine-tuned on MIMIC-Data that significantly improves both user-like cognitive reasoning and human-like comment expression. The reviewers recognized the significance of MIMIC-Bench, highlighting its clear motivation, timely relevance, comprehensive task design, strong presentation quality, and the importance of evaluating human-like cognitive and social reasoning in multimodal video understanding. Although reviewers raised several concerns regarding the design, difficulty, and evaluation of MIMIC-Bench and MIMIC-Data, most of these concerns are successfully addressed in the rebuttal. The manuscript is also updated accordingly. Overall, the rebuttal substantially improves clarity and completeness.
** This paper is conditionally accepted provided the authors do the following for the camera-ready**:
[Ethics] Authors must describe in detail steps taken to ensure that they respect copyright and terms of service, that released data does not contain identifiable information and to ensure that the released data cannot be used to cause reputational harm.

**Reviewer Concerns:**

The reviewer's concerns are organized as follows:

W1: The difficulty MIMIC-Bench (Reviewer Y9c1) [in the middle]

R1: The authors successfully argue that MIMIC-Bench is not trivially easy overall and has discriminative power. However, they do not fully resolve concerns about benchmark depth and case difficulty as well as distractors.

W2: Does not report the video length distribution of MIMIC-Data (Reviewer Y9c1) [addressed by the rebuttal]

R2: The authors reported this information during rebuttal.

W3: Data separation and overfitting concerns of MIMIC-Data (Reviewer Y9c1 & cT9q) [addressed by the rebuttal]

R3: The concern about overfitting and leakage is addressed with clear procedural guarantees and new experimental evidence during rebuttal.

W4: Clarify the annotation process, bias mitigation, and consistency (Reviewer cT9q) [addressed by the rebuttal]

R4: The authors clarified this point during rebuttal.

W5: Failure modes and cross-benchmark/OOD generalization (Reviewer cT9q & wUW5) [addressed by the rebuttal]

R5: The concern about engagement bias, lack of diversity, and missing diagnostic analysis is largely addressed through new experiments and clearer analysis.

W6: Clarify the evaluation procedure of the user-like mimicking task (Reviewer f8nN & wUW5) [addressed by the rebuttal]

R6: The authors clarified this point during rebuttal.

**Reviewer Scores:**

Reviewer Y9c1: 2 -> 4 (concerns are partially addressed, raises the score)

Reviewer f8nN: 6 -> 6 (all concerns are addressed, remains positive rating)

Reviewer cT9q: 4 -> 6 (all concerns are addressed, raises the score)

Reviewer wUW5: 8 -> 8 (all concerns are addressed, remains positive rating)

Average score: 6

---

### Decision · Program_Chairs · 2026-01-26

**Decision:**

Accept (Poster)

**Comment:**

Conditions for acceptance have been satisfied.